# High-performance cost efficient simultaneous wireless information and power transfers deploying jointly modulated amplifying programmable metasurface

Xin Wang[1], Jia Qi Han[1], Guan Xuan Li[1], De Xiao Xia[1], Ming Yang Chang[1], Xiang Jin Ma[1], Hao Xue[1], Peng Xu[1], Rui Jie Li[1], Kun Yi Zhang[1], Hai Xia Liu[1], Long Li [1] ✉ & Tie Jun Cui [2] ✉

Programmable metasurfaces present significant capabilities in manipulating electromagnetic waves, making them a promising candidate for simultaneous wireless information and power transfer (SWIPT), which has the potential to enable sustainable wireless communication in complex electromagnetic environments. However, challenges remain in terms of maximum power transmission distance and stable phase manipulation with high-power scattered waves. Additionally, waveform limitations restrict average scattered power and rectifier conversion efficiency, affecting data transmission rates and energy transmission distance. Here we show an amplifying programmable metasurface (APM) and a joint modulation method to address these challenges. The APM mitigates the peak-to-average power ratio and improves maximum power, phase response stability, average output power, and rectifier conversion efficiency. Through experimental validation, we demonstrate the feasibility of the SWIPT system, showcasing simultaneous LED array powering and movie video transmission. This innovative SWIPT system holds promise for diverse applications, including 6 G wireless communications, IoT, implanted devices, and cognitive radio networks.

With the rapid growth of wireless networks, there is a huge demand for sustainable power and information exchange for the enormous number of low-power devices and wireless sensors. Simultaneous wireless information and power transfer (SWIPT) is considered a promising solution to solve the bottleneck problems of low-power devices and wireless sensors in perpetual communications[1,2]. In contrast to conventional wireless information transfer (WIT) systems[3] and wireless power transfer (WPT) systems[4], the SWIPT systems are focused on balancing the communication rate and the amount of delivered energy to achieve the optimal rate-energy transfer[5,6]. In general, the

communication rate increases with modulation complexity. For example, 1024-QAM (Quadrature amplitude modulation) can achieve five times the data rate of 4-QAM at the same symbol rate. However, the high-order modulation typically leads to a high peak-to-average power ratio (PAPR), which is defined as the ratio of the maximum power to the average power. This high PAPR can significantly impact the performance of the SWIPT system in two ways. Firstly, the instantaneous peak signal with high PAPR can exceed the breakdown voltage of the diode in the rectifier, which causes a notable deterioration in the maximum conversion efficiency of rectifier[7,8].

[1]Key Laboratory of High-Speed Circuit Design and EMC of Ministry of Education, School of Electronic Engineering, Xidian University, Xi'an 710071, China. [2]Institute of Electromagnetic Space and the State Key Laboratory of Millimeter Waves, Southeast University, Nanjing 210096, China. ✉e-mail: lilong@mail.xidian.edu.cn; tjcui@seu.edu.cn

Secondly, the peak transmitted power is usually restricted by the saturation output power of the transmitter, resulting in low output average power under high PAPR signals[9]. As the fundamental of the SWIPT system, the transmitter is crucial to determine the system's performance and architecture. Generally, multi-antenna systems are introduced into conceptual schemes for the transmission of information and energy[10,11]. However, the multiple-antenna systems require many transceivers and radio-link circuits, which are expensive, bulky, and inefficient in wireless power transfers[12,13]. The phased array system is a traditional solution for the near-field and far-field WPTs due to its flexible beamforming capability[14,15]. Unfortunately, phased array systems require many transceiver modules and complex feeding networks, making them bulky, costly, and difficult to fabricate.

Recently, reconfigurable intelligent surface (RIS) has been considered a promising solution for SWIPT due to its extraordinary ability in wavefront manipulations[16,17]. As the basic hardware structure of RIS, the programmable metasurface (PMS) demonstrates the powerful ability to modulate electromagnetic waves in real time[18–26]. Generally, PMS comprises artificial subwavelength structures and electrically lumped components such as positive-intrinsic-negative (PIN) diodes, varactors, and micro-electro-mechanical systems (MEMS). By dynamically controlling the lumped components via field-programmable gate array (FPGA), PMS can realize various interesting functions in tailoring the EM waves, such as beam scanning[27–30], vortex beam generation[31–33], energy focusing[34,35], and polarization editing[36–38]. Despite the excellent capabilities of PMS for EM wave manipulations, the scattering intensity of the unit cell restricts its application in charging wireless devices. The low power tolerance and scattering intensity are mainly attributed to the lossy components (PIN diodes, varactor diodes, and MEMS) embedded in the unit cells of PMS. We termed the PMS integrated with lossy components as lossy PMS because it will attenuate the incident wave intensity. On the other hand, the lossy PMS needs a large aperture to focus the incident waves to the specific area in a short range[39], severely limiting its applications in practical SWIPT systems.

Compared to lossy PMS, amplifier-based metasurfaces can amplify spatial waves, which can significantly reduce the aperture of PMS and effectively compensate for the path loss from the access point[40,41]. In addition to enhancing the spatial waves, embedding the amplifier in the unit cell could introduce various remarkable physical properties such as nonreciprocal scattering[42–47], spatial frequency multiplication[48], and reflection enhancement[49,50]. For the SWIPT application, the transmitted energy is much larger than the information due to different power sensitivities in the rectifier and demodulator, resulting in the high emitted power of the transmitter. So far, the reported amplifier-based metasurfaces only amplify the incident waves at a low energy level, which is unsuitable as a transmitter for the SWIPT systems. Besides the scattering power, stable phase manipulations of the amplifier-based metasurfaces in high-power scenarios are of great importance for forming high-quality energy beams, which is the primary challenge in the design of amplifier-based metasurfaces. In addition, both PMS and amplifier-based metasurfaces are scattering structures whose maximum and average scattering powers are also limited by the envelope of the incident signal.

In this paper, we propose a jointly modulated amplifying programmable metasurface (APM) as a SWIPT transmitter to transmit information and energy simultaneously. The APM consists of periodical amplifying unit cells to increase the maximum power of reflected waves. Each amplifying unit cell can be decomposed into four parts, which can manipulate the incident waves with four discrete states as detailed in Supplementary Note 4. With dynamic manipulations of the phase distribution on APM, we could combine energy and information in a highly concentrated beam similar to a single beam for flexible wireless charging and data transmission. According to the plane-wave angular spectrum method, we could readily calculate the beam scanning of APM and validate the required phase distribution on the surface. In addition, we present a joint modulation strategy to effectively mitigate the degradation caused by high-order digital modulations by reducing PAPR. This method significantly improves the rectifier's conversion efficiency and the average output power of APM. We construct a SWIPT system to verify the feasibility of the APM and the joint modulation strategy. Compared to the state-of-the-art PMS[34,39,51,52] and amplifier-based metasurfaces[37,40–42,44,45,49,50], the proposed APM system improves the maximum transmitted power and flexibility in conveying energy and information, which has potential applications in wireless implantable devices, 6 G communication networks, and the Internet-of-things (IoT).

## Results
### Jointly modulated APM for SWIPT
The proposed jointly modulated APM simultaneously transmits electromagnetic (EM) energy and digital information to improve the sustainability of wireless communication, as shown in Fig. 1a. In contrast to the conventional SWIPT system, the jointly modulated APM could enhance and form the beams to the terminal by using the APM and joint modulation strategy. The joint modulation strategy combines digital signals and continuous-wave (CW) signals at different frequencies in a highly concentrated beam, which can reduce the transmitted signals' PAPR and improve APM's outgoing-wave power and the conversion efficiency of the rectifier circuit. The basic principle of joint modulation is detailed in the method and the theoretical analysis of PAPR on the conversion efficiency, as shown in Supplementary Note 3. Since 2-bit coding with four distinct phase states is the optimal tradeoff solution between the energy loss of the steering beam caused by phase quantization and the power consumption as well as the complexity and cost of the APM hardware, here we choose the 2-bit solution for regulating the reflected beam. The analysis of phase quantization is detailed in Supplementary Note 8. The designed APM consists of $M \times N$ amplifying unit cells, which can dynamically generate the required high-power beams by controlling the phase distribution using the field-programmable gate array (FPGA). As the fundamental part of APM, the amplifying unit cell consists of an amplifier circuit, a reradiating patch, a receiving patch, and a reconfigurable phase shifter. Moreover, full-wave simulation results in HFSS of ANSYS Desktop 2020 demonstrate that the amplifying unit cell can enhance the reflected-wave intensity with four distinct states (Fig. 1b(i)). Phase responses are quantized to four states (Fig. 1b(ii)), indicating that the APM can manipulate the direction of outgoing waves. The design and validation of the amplifying unit cell are detailed in Supplementary Note 4.

### Real-time wireless information and power transmitting
APM is the basic hardware to steer beams to simultaneously transmit the video signals and electromagnetic power to the receiving antenna, as shown in Fig. 2a. In contrast to asynchronous SWIPT systems based on time and space switching technology[2,10,53], the energy and information are carried by the highly concentrated beam and delivered to the same terminal in APM system. That means the terminal could obtain the information and power at the same time. Here, we conduct two experiments to verify the feasibility and performance of the proposed system. One experiment is lighting up LED arrays and transmitting 4QAM data in different directions (see Supplementary Movie 1). The other is transmitting real-time video and lighting up LED arrays (see Supplementary Movie 2). The joint modulation strategy combines CW and digital signals at different frequencies. Here, we select a 4 GHz CW signal and a 4QAM modulation signal with a bandwidth of 2 MHz centered at 3.995 GHz to carry EM energy and video information, respectively. From the spectrum of joint modulation shown in Fig. 2b, the power ratio of the CW signal to the 4QAM signal is 30 dB to reduce the PAPR of the transmitted signal. At 2 m apart from

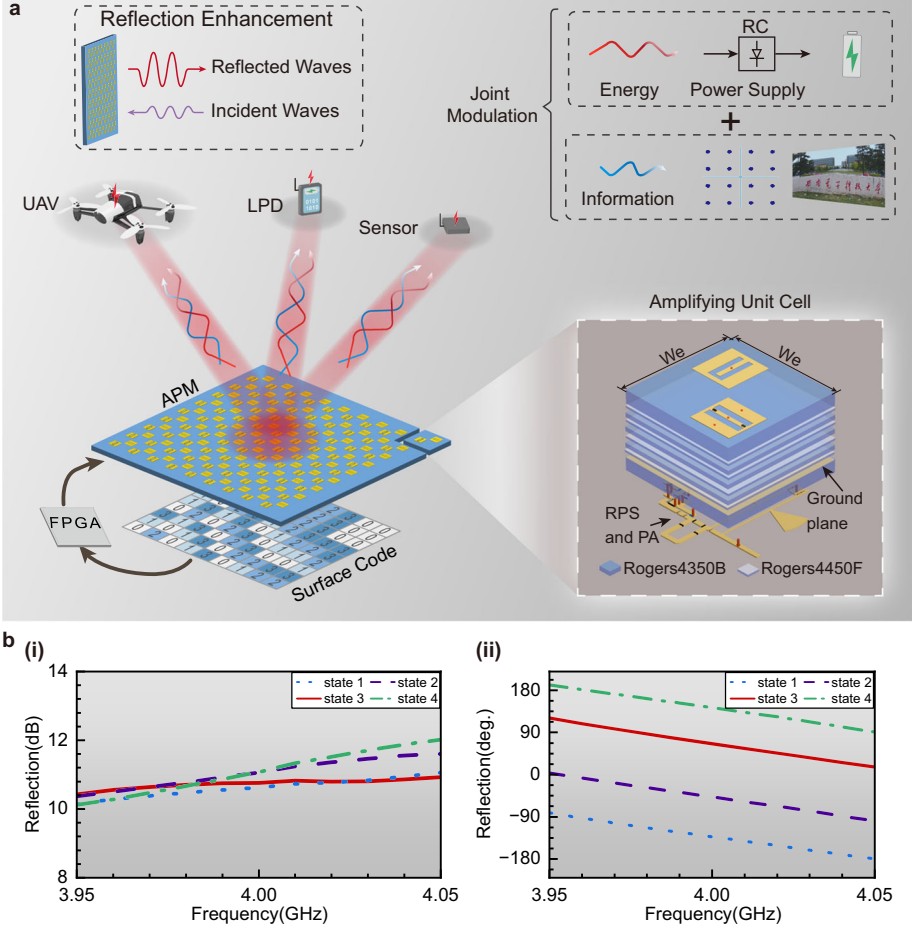

**Fig. 1 | Concept view of jointly modulated APM system for SWIPT. a** Architecture of system based on jointly modulated amplifying programmable metasurface (APM), which could simultaneously transmit the wireless energy and information to the unmanned aerial vehicle (UAV), low-power device (LPD), and sensors. Inset: The upper-right inset is the joint modulation that combines the energy and information signals. The energy source converted by a rectifier circuit (RC) charges the terminal devices; the upper-left inset represents the reflection enhancement by the APM; the bottom-right inset illustrates the explosion view of the amplifying unit cell which consists of two patches, a reconfigurable phase shifter (RPS), and a power amplifier (PA). **b** The simulation results of the amplifying unit cell under periodical boundary conditions, including the (**i**) magnitude and (**ii**) phase of the reflection.

APM, we obtained the converted DC power and output DC voltage by employing the rectifier circuit as depicted in the inset of Fig. 2c. The design of the rectifier circuit is detailed in Supplementary Note 6. From Fig. 2c, the converted DC power is more than 16 mW, and the output DC voltage is more than 1.75 V, implying that APM could power thousands of LPDs at the same time (an LPD only needs uW-level power consumption[54]). The converted DC power can be increased until it reaches the maximum output power of APM. We also validate the wireless communication performance of APM system. Bit-Error-Ratio (BER) of the transmitted signals are less than $10^{-4}$ in the coverage area of APM, indicating good quality of the data transmissions over a wide range from −30° to 30°, as shown in Fig. 2d. Transmitted video are demodulated and played in laptop shown in Fig. 2e, f, which demonstrates the reliability of wireless communication of APM system in SWIPT application.

It is worth noting that the relative frequency difference between the energy and information can be flexibly configured in the operating band from 3.95 GHz to 4.05 GHz. The relative frequency difference between information and energy primarily depends on the PAPR of the transmitted signals, the quality of the data transmission, and the beam performance. According to the joint modulation method, the PAPR can be manipulated by adjusting the ratio of CW signal to the information signal. The amplitudes of CW and information can be independently controlled at different frequencies, which is independent of the relative frequency difference. In terms of data transmission quality, the proposed APM can maintain high-quality data transmission because it enhances the signal energy in the operating frequency band instead of attenuating it. Additionally, measurement results show that the deviations in maximum output power, half-power beamwidth, and beam direction are 0.71 dB, 1.78°, and 3°, respectively, which guarantee the flexibility for the energy and information configuration on frequency. The deviation in beam performance is detailed in Supplementary Note 7.

The signals with a high peak-to-average power ratio (PAPR) will reduce the average output power in terms of the saturated reflected power of APM and deteriorate the conversion efficiency of the rectifier in a wide input power range. We construct a simple QAM architecture which is combined with a continuous wave signal $A_p \cos(2\pi f_p t)$ as a schematic of joint modulation for signal performance analysis, as shown in Fig. 3a. The schematic of joint modulation can be reduced to the conventional QAM architecture when $A_p = 0$. The square root cosine filter (SRC) is employed as the pulse-shaping filter to eliminate intersymbol interference(ISI), and the roll-off factor is assumed to be 0.5. Additive white Gaussian noise(AWGN) is inserted before SRC filter to evaluate the transmitted signal without loss of generality. According to communication architecture (Fig. 3a), the envelope of the conventional signal and joint modulation signal is readily achieved with the assumption of $A_p = 0$ and $A_p = 0.05$, as depicted in Fig. 3b. From the

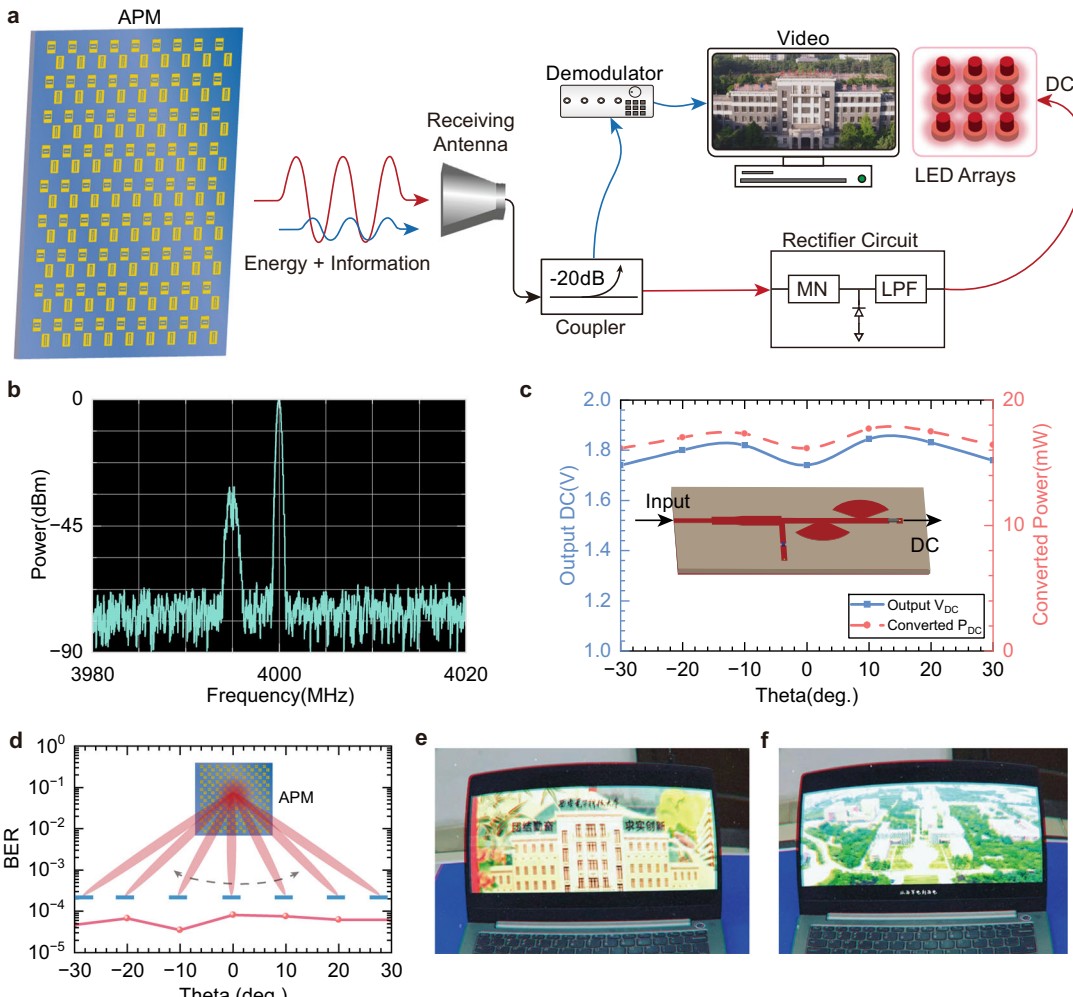

**Fig. 2 | Experimental validation of jointly modulated APM system. a** Jointly modulated amplifying programmable metasurface (APM) simultaneously transmits video signals and EM energy in a single beam to the terminal. A rectifier circuit consists of a matching network (MN) and a low pass filter (LPF), outputting direct current (DC) to the light-emitting diode (LED) arrays. **b** Measured spectrum of the received signals from the receiving antenna. **c** Measured output DC voltages $V_{DC}$ and converted DC power $P_{DC}$ of the rectifier at steering angles of APM from −30° to 30°. **d** Measured Bit-Error-Ratio (BER) of the receiving signals at different scanning angles of APM. The receiving antenna is 2 meters apart from the APM. **e**, **f** Demodulated video from the coupler port.

envelope results of signals, we found that the joint modulation signals have much lower PAPR than conventional signals. In other words, joint modulation can significantly mitigate the effect of the digital modulation signal on the PAPR, enabling complex high-order modulation and further improving signal transmission rate and DC output power.

Moreover, we investigate the PAPR performance by considering the proportion of CW to QAM signals (Fig. 3c). We found that the PAPR drops sharply with the proportion of CW signal increase. Generally, the probability of the instantaneous power of a signal can be described by a complementary cumulative distribution function. It is a statistical method to analyze the instantaneous power of the transmitted signal above its average power, corresponding to the PAPR distribution. The distribution of PAPR of different-order modulated signals is measured by a signal analyzer 4051 manufactured by Ceyear company, as depicted in Fig. 3d, verifying that the joint modulation has much lower PAPR than conventional modulation signals. Notably, we measured the PAPR values of all modulated signals in Fig. 3d under the same average output power 10 dBm. The ratio of CW and 64QAM signals in the joint modulation method is 30 dB in Fig. 3d, e. For wireless information transmission, the power of the information does not need to be very large because the receiver has good sensitivity to demodulate the input signal. In addition, the CW is orthogonal to the modulated

baseband signal, resulting in the signal-to-noise ratio (SNR) of the baseband signal being independent to the CW signal, which is verified by the experiment shown in Supplementary Movie 3. The SNR varies less than 2.84 dB during the experimental process, demonstrating that the proposed APM and joint modulation method can realize reliable and high-quality data transmissions, as detailed in Supplementary Note 11. In order to verify the effect of modulated signals with different PAPR values on the rectifier circuit, we design and fabricate a single-diode rectifier circuit (Supplementary Note 6). The measured conversion efficiency of the rectifier circuit is the ratio of the output DC energy to the input RF power, which obviously decreases as the conventional modulation order increases, as illustrated in Fig. 3e. This is because a higher-order modulation will result in the transient peak signals with higher PAPR exceeding the cutoff region of the Schottky diode. It is worth noting that the measured joint-modulation signal has a similar conversion efficiency to the CW signal, verifying that the joint-modulation method can significantly mitigate the effect of conventional high-order modulated signals.

**Enhancements of spatial waves**
The proposed APM can enhance the spatial wave rather than attenuate it. To verify the enhancement of APM, we measured the intensity of the

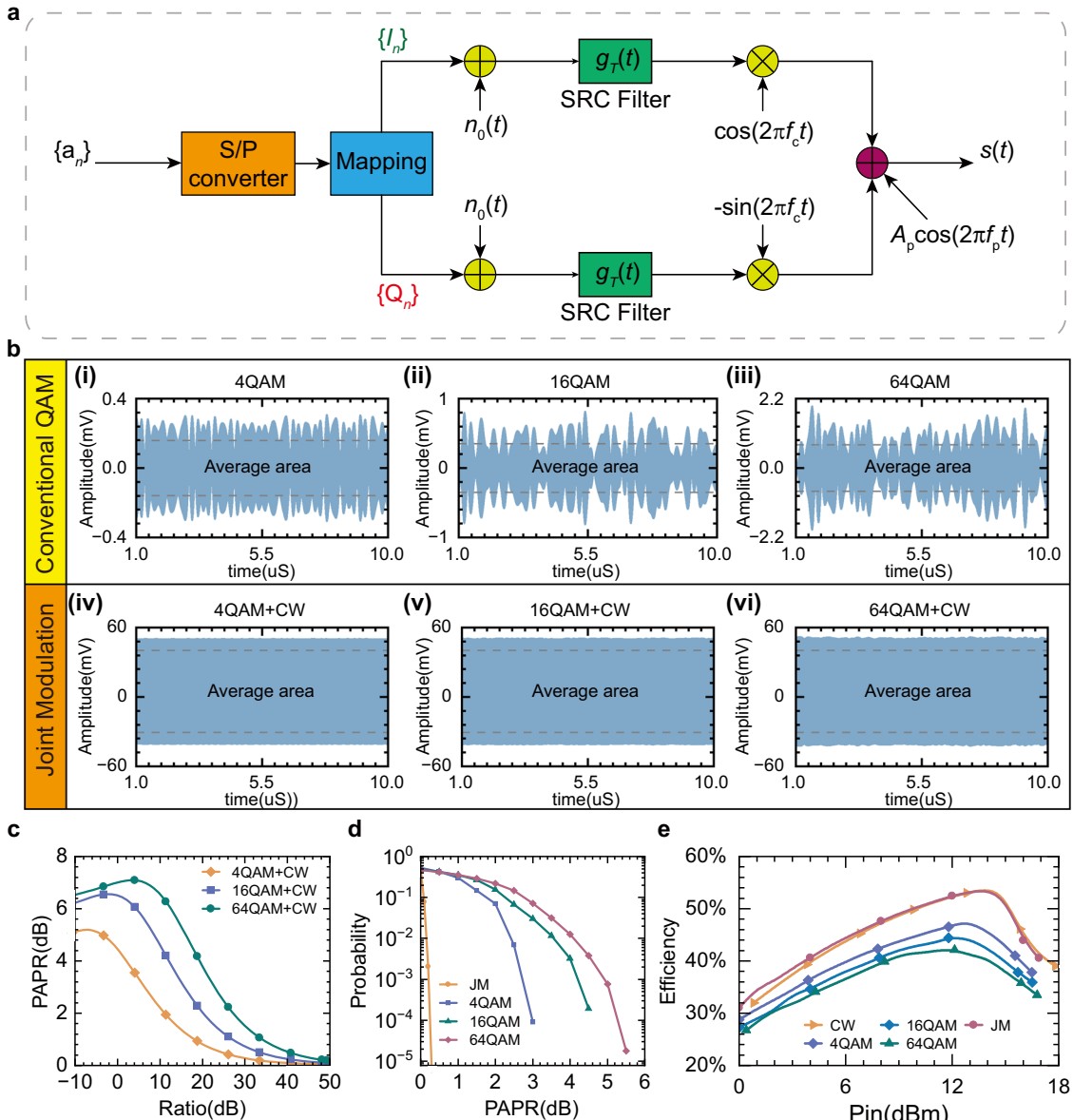

**Fig. 3 | Joint modulation analysis and experimental results. a** Schematic of joint modulation for analyzing the signal performance. A serial-to-parallel converter converts random signals to I and Q channels. Two square root cosine (SRC) filters are inserted before mixers to eliminate intersymbol interference. **b** Theoretical comparison of the signal envelope between conventional quadrature amplitude modulation (QAM) and joint modulation (JM) signals. Three examples of conventional QAM signals, such as (**i**) 4QAM signal, (**ii**) 16QAM signal, and (**iii**) 64QAM

signal. The envelope (**iv**), (**v**), and (**vi**) of JM signals is the combination of three different QAM signals with continuous wave signals. The average area is the average of envelopes of each modulated signal. **c** The peak-to-average power ratio (PAPR) under different power ratios of continuous wave (CW) signal to QAM signal. **d** Measured probability of different modulation signals versus PAPR. **e** The measured conversion efficiency of the rectifier in different signals versus the input power.

reflected wave in a microwave chamber. The schematic of the experimental setup is depicted in Fig. 4a. Two standard horn antennas serve as the receiving and feeding antennas to generate and acquire orthogonally polarized waves. The receiving antenna is orthogonal to the feeding antenna to acquire the *y*-polarized waves radiated from APM. When the incident *x*-polarized wave impinges on APM, it will reflect the enhanced *y*-polarized waves in the predicted direction. Receiving and feeding antennas are connected to the vector network analyzer (VNA) ports to record the scattering pattern.

According to the measured scattering patterns in Fig. 4b, four scattering beams (red lines) pointing at 0°, 10°, 20°, and 30°, respectively, indicating that APM can achieve dynamic manipulations of waves by adjusting the phase distribution using the FPGA circuit. When exchanging the feeding and receiving antenna's polarization, we

obtain another scattering pattern (blue lines), implying that the prototype only reflects and enhances the *x*-polarized incident waves. The upper and lower bounds in Fig. 4c are the scattering patterns of the APM and the lossy metasurface, respectively. The structure of the lossy element surface is the same as that of the APM, except that the power amplifier circuit is replaced by a microstrip line. The area is the enhancement of APM compared with lossy metasurface. The enhancement of APM to spatial waves could significantly reduce the aperture of the transmitter and compensate for the path loss from the source.

### The maximum reflected powers of APM
The maximum reflected power directly determines the power charging coverage of APM in the practical system, which depends on the

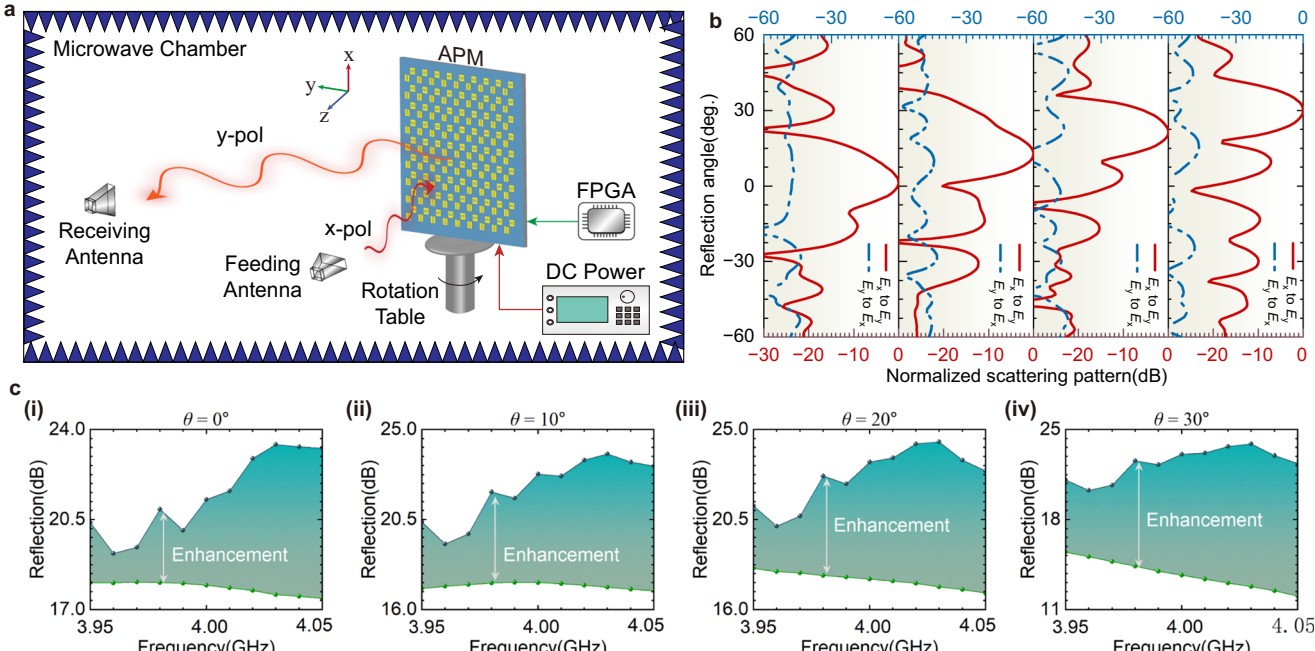

**Fig. 4 | Reflection enhancements and experimental results. a** Schematic of the experimental setup to evaluate the enhancement of amplifying programmable metasurface (APM). A field programmable gate array (FPGA) is utilized to control the phase distribution of APM. A direct current (DC) power supply provides DC energy to APM. **b** Measured scattering patterns of APM ranging from 0 to 30°, including the $x$-polarized wave ($E_x$) to $y$-polarized wave ($E_y$)reflections (red lines) and the $y$-polarized wave to $x$-polarized wave reflections (blue lines). **c** Scattering enhancements in the operating bandwidth at different steering angles, such as **(i)** $\theta = 0°$, **(ii)** $\theta = 10°$, **(iii)** $\theta = 20°$ and **(iv)** $\theta = 30°$.

maximum reflected power of the unit cell. The maximum reflected power of the unit cell is over 27 dBm according to the load-pull analysis, as detailed in Supplementary Note 4. A schematic view of the experimental setup for the maximum reflected power evaluation is depicted in Fig. 5a. By adjusting the distance and angle of the receiving antenna relative to APM, we recorded the maximum output powers at different distances and scan angles. For verifying the relationship between the phase distribution of APM and steering angle, the plane-wave angular spectrum (PWAS) method is utilized (Supplementary Note 1). For instance, three different phase distributions are used for beam generation with three different steering angles, as shown in Fig. 5b. In turn, the field distribution based on the PWAS method confirms the feasibility of the phase distribution.

From Fig. 5c, we observe that over 30 dBm could be achieved at 1 meter apart from the boresight of the prototype, which verifies that APM can provide enough energy to terminals. When the wireless energy transmission distance increases from 1 to 3 meters, the receiving antenna could capture more than 24 dBm from APM. For evaluating the wireless coverage of APM, the maximum reflected power is recorded at different positions, as depicted in Fig. 5d. Over 23.5 dBm is received by receiving antenna in 60° coverage at 2 meters apart from APM, which verifies that the amplifying unit cell could enlarge the maximum power of the system. The design and verification of the amplifying unit cell are detailed in Supplementary Note 4 and Note 5.

## Discussion

We propose a jointly modulated APM to realize simultaneous transmissions of information and energy. In contrast to the MIMO techniques introduced in the conventional SWIPT systems[2,10,53], APM provides much more flexibility in manipulating the electromagnetic beam and reduces the number of radio link circuits. Therefore, the APM system can significantly reduce the complexity, size, and cost of the SWIPT system. Since the joint modulation strategy combines the

energy and information in a highly concentrated beam similar to a single beam, the APM can simultaneously charge and communicate with the terminal. By leveraging the independent modulations of energy and information and the stable physical channel with minimal spatial variation, it becomes feasible to flexibly configure the frequencies of both energy and information in the operating band. A stable physical channel in the operating band is guaranteed by generating a highly concentrated beam similar to a single beam by APM (see Supplementary Note 7). Unlike the time-switching or space-switching techniques reported in other SWIPT system[53], the proposed jointly modulated APM improves the output DC power with ultra-low PAPR, notably boosting the conversion efficiency and coverage. Moreover, the amplifying unit cell not only enhances the intensity of the spatial wave but also enlarges the maximum outgoing power, which further increases the wireless charging and communication distance and reduces the aperture size of the metasurface. Two indoor experiments were performed to validate the concept and performance of the proposed APM system (Supplementary Fig. 11). One is to use a 4QAM signal and CW signal as joint modulation for information and energy transmission (Supplementary Movie 1). The other is to transmit video and light up the LED array at the same time (Supplementary Movie 2). Experimental results demonstrate the feasibility of the jointly modulated APM system, indicating that the joint modulation strategy can significantly mitigate the effect of the modulated information. Although the experiments were performed within 3 meters, the far-field wireless information and energy transmission can be achieved by increasing the input power of APM.

The benefits of the proposed joint-modulated APM for SWIPT are limited by the performance of APM, such as the level of enhancement, operating bandwidth, and the maximum output power from APM. To address these limitations, future improvements could involve implementing a wideband power amplifier and patch scatterer to enhance the operating band. In addition, utilizing an amplifier with higher gain and saturation output power or enlarging the aperture of the surface

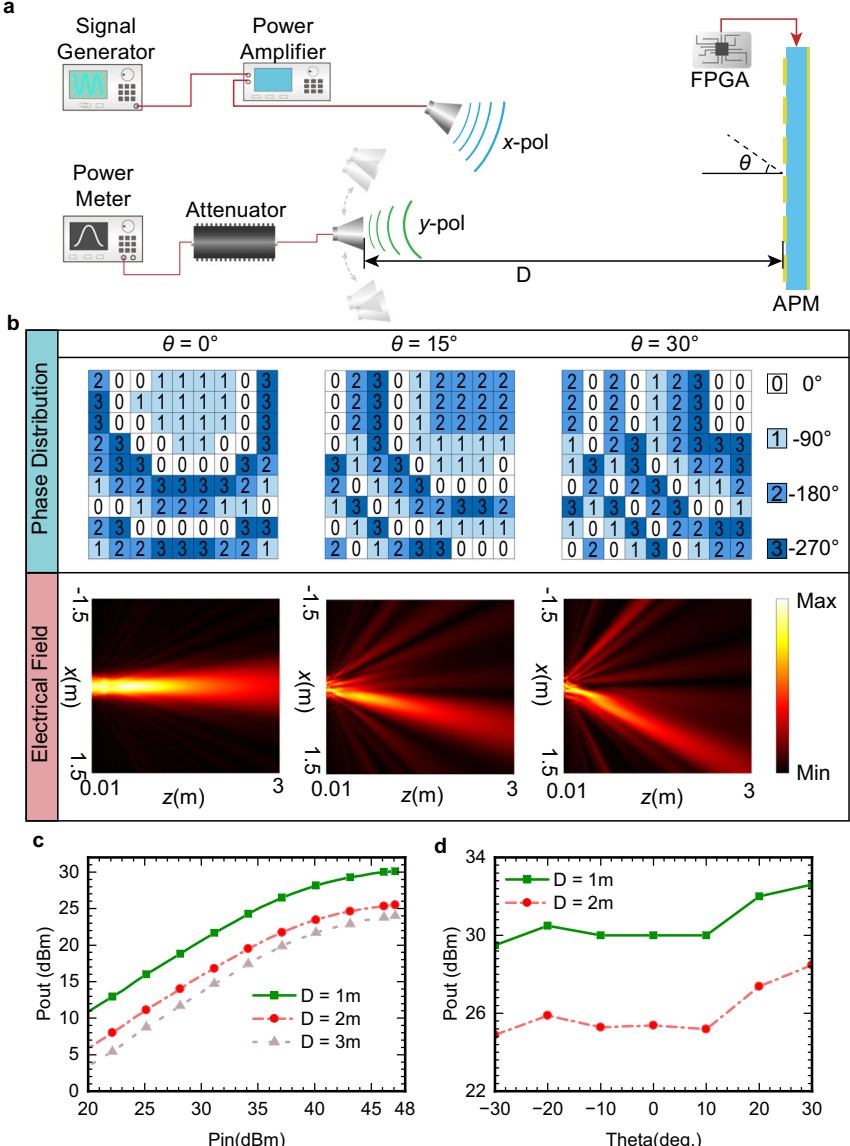

**Fig. 5 | Experimental results of maximum reflected power of APM. a** Schematic of the experimental setup for saturated power evaluation. A field programmable gate array (FPGA) is utilized to control the phase distribution of amplifying programmable metasurface (APM) for reconfiguring high-directional beams. $\theta$ is the angle between the outgoing beam and the normal line of APM. D is the distance between the receiving antenna and APM. **b** Synthesized electric field distribution using different phase distributions on the APM at steering angle $\theta = 0°$, $\theta = 15°$, and $\theta = 30°$. **c** Measured reflected powers along the boresight of APM at different input powers and different distances. **d** Maximum reflected powers at different scanning angles of APM with different distances.

would prove beneficial in increasing the maximum output power of APM, thereby extending the range for wireless charging and information transmission.

## Methods
### Design of the APM
The APM is designed using multilayer printed circuit board (PCB) technology and has the dimensions of 450 mm × 500 mm × 4.204 mm, consisting of 9×9 amplifying unit cells and a bias network. The green part shown in Supplementary Fig. 10b is the bias network connected with each unit cell's PIN diodes and power amplifiers. We use two 110-pin high-speed connectors to connect the prototype and FPGA board to program the phase distribution of APM. Two power connectors on the bottom-right of Supplementary Fig. 10b(ii) are connected to a DC power supply to provide +5 V voltage to each unit cell's amplifier. An orthomode transducer waveguide is designed and applied to verify the performance of the unit cell (Supplementary Note 5). From the

waveguide measurement, the maximum reflected power of the unit cell is 27 dBm, and the variation of four discrete phase responses as the input power is less than 25.5°, which indicates good phase stability and power capacities of the unit cell for SWIPT.

### Joint modulation strategy
The underlying theory of joint modulation can be expressed as follows. The total harvested energy is a sum of baseband and continuous waves, denoted by

$$P_T = \eta\left(P_i, PAPR\right)\left(E\left[\left\|\mathbf{H}\mathbf{x}(n)\right\|^2\right] + P_c\right) \tag{1}$$

And the baseband transmission from the APM to the receiver can be modeled as

$$\mathbf{y}(n) = \mathbf{H}\mathbf{x}(n) + z(n) \tag{2}$$

Since only one physical channel in the proposed system uses a high-directional beam and one antenna equipped on the receiver ($M = 1$, $N = 1$), the total harvested energy can be written as

$$P_T = \eta(P_a, PAPR)\left(H_b(\theta, \varphi)E\left[||\mathbf{x}(n)||^2\right] + H_c(\theta, \varphi)P_e\right) \quad (3)$$

We suppose that the ratio of the continuous wave to the baseband signal energy is $\alpha$. Hence, we can adjust $\alpha$ to improve the total harvested energy while keeping the baseband energy for high-quality data transmission. Furthermore, the PAPR of the total transmitted signal can be readily reduced by increasing the ratio $\alpha$. According to the field calculation in equation S1 in Supplementary Note 1, the transmission coefficient in the Fresnel Zone is proportional to the scattered energy $H(\theta, \varphi) \propto |E_y|^2$. Hence, we can adjust the code distribution on the APM to form an arbitrary high-directional beam for energy and information transmission. The more detailed discussion about joint modulation and its effect on conversion efficiency is presented in Supplementary Note 2 and Supplementary Note 3.

For conceptual clarity, a simple QAM modulation architecture with additive white Gaussian noise is shown in Fig. 3a. Square root raised cosine filters with a roll-off factor of 0.5 are inserted before QAM modulator in the transmit channel as an adaptive filter to eliminate the intersymbol interference. After the shaping filter, the symbol rate is 5 MHz to generate a 7.5 MHz bandwidth signal. The joint modulation signal can transmit wireless energy and digital information by combining the CW at 4 GHz with the modulated 16 QAM signal centered at 3.95 GHz. Increasing the proportion of CWs in the joint signal will reduce PAPR of the transmitted signal and improve the efficiency of the rectifier circuit. It is worth noting that the ratio of CW signal to QAM signal in the joint signal is 20 dB. From the calculated results, the joint modulation method notably reduces the PAPR. In order to accurately observe the transmitted signal, we set the sampling rate to 20 GHz in the calculation.

## Experimental validation

We conducted three experiments in the microwave chamber and indoor environments to evaluate the prototype's performance. The measurement in the microwave chamber mainly contains a feeding standard horn antenna LB-284-10 from INFO company, a receiving standard horn antenna LB-20180 from INFO company, and a vector network analyzer N5244A from KEYSIGH company for far-field intensity evaluation of APM (Supplementary Fig. 10a). In the far-field measurement, the receiving horn records the outgoing wave intensity from the APM. The APM is fixed on a rotation table to record the scattering electrical field with different angles. The feeding antenna has an orthogonal polarization with the receiving horn. We recorded the output power from a standard horn antenna WR284 as a reference to calculate the scattering pattern of APM.

For the indoor experiments, there are two measurement setups in the hallway to verify the SWIPT of the APM system. The measurement in the hallway contains two standard horn antennas LB-284-10, a 20 dB coupler, a 3 dB combiner, a rectifier, a signal generator, a power amplifier, a DC power supply, two software-defined radios, and a laptop. We combine the modulated baseband signals with continuous waves by a 3 dB combiner. The modulated baseband signal and continuous wave signal are generated by an SDR and a signal generator, respectively. One hallway experiment shows that the proposed APM can transmit information and wireless energy at different locations, as shown in Supplementary Movie 1. We reconfigure the steering angle of APM by digital control circuit. A 4QAM baseband signal is transmitted to demonstrate the information transmission. The power ratio between the modulated baseband signal and the CW signal is 30 dB. The distance between the receiving horn antenna and APM is 2 meters. Here, we select a 4 GHz CW signal and a 4QAM modulation signal with a bandwidth of 2 MHz centered at 3.995 GHz to carry EM energy and

digital information, respectively. When the receiving antenna is located within the beam of the APM, the LED array is illuminated and the quality of information transmission is best. In another hallway experiment, a video signal and energy are transmitted simultaneously, as shown in Supplementary Movie 2. In this experiment, the receiving antenna is 2 meters away from the boresight of APM. The video signal is modulated by using 4QAM with 2 MHz bandwidth centered at 3.995 GHz. A 4 GHz CW signal is utilized to carry wireless energy.

## Data availability

All other data are available from the corresponding authors on request. The experimental and simulated data of figures in the main text are provided in the Source Data file. Source data are provided with this paper.

## Code availability

Code used in this work is available from the corresponding authors on request.

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

## Acknowledgements

The authors acknowledge the National Natural Science Foundation of China (No. 62288101), National Key Research and Development Program of China under Grant No. 2021YFA1401001, Key Research and Development Program of Shaanxi (No. 2021TD-07), and the Fundamental Research Funds for the Central Universities.

## Author contributions

L.L. and T.J.C. suggested the designs, planned and supervised the work, and in consultation with X.H.L., X.W. conceived the idea, and carried out the analytical modeling, numerical simulations, and sample fabrication.

J.Q.H., D.X.X., X.J.M., P.X., H.X., and K.Y.Z. performed the data analysis and measurements. G.X.L. prepared the FPGA code and digital hardware. M.Y.C. designed the rectifier circuit. R.J.L. designed the supporting materials using 3D printing technology. All authors discussed the theoretical aspects and numerical simulations, interpreted the results, and reviewed the manuscript.

## Competing interests

The authors declare no competing interests.
