## [Peer review file · Nature Communications]

REVIEWER COMMENTS

Reviewer #1 (Remarks to the Author):

The concept of simultaneous wireless information and power transfers (SWIPT) is not new and it has been studied by various academic communities for a long time since 2008. There is a long list of papers can be found below:

"Simultaneous Wireless Information and Power Transfer in Modern Communication Systems" by Zhang, R., & Ho, C. K. (2013)

"Simultaneous Wireless Information and Power Transfer: A Survey of Recent Developments" by Ju, H., Zhang, R., & Guizani, M. (2014)

"Simultaneous Information and Power Transfer for Energy Harvesting Communication Systems" by Varshney, L. R. (2008)

"Simultaneous Wireless Information and Power Transfer in Cooperative Networks" by Liu, L., Zhang, R., & Chua, K. C. (2013)

"Simultaneous Wireless Information and Power Transfer in MIMO Systems" by Clerckx, B., & Oechtering, T. J. (2013)

"Simultaneous Information and Power Transfer for Broadband Wireless Systems" by Park, J. M., Zhang, R., & Fung, C. (2013)

"Simultaneous Information and Power Transfer in Multiuser MIMO Systems" by Zhou, X., Zhang, R., & Nallanathan, A. (2013)

"Simultaneous Wireless Information and Power Transfer in Cognitive Radio Systems" by Dey, S., Niyato, D., & Kim, D. I. (2013)

"Simultaneous Wireless Information and Power Transfer for MIMO-OFDM Systems" by Huang, K., Wu, D., & Zhang, R. (2014)

"Simultaneous Information and Power Transfer in Single-Antenna Interference Networks" by Tandon, A., & Motani, M. (2014)

The paper combined this concept with programmable surfaces or digital coding surfaces, it however does not address what the added value is for SWIPT. There are ongoing challenges such as improving power efficiency, which involves optimizing the energy transfer efficiency and reducing energy losses during transmission, conversion, and harvesting processes. Balancing the power requirements with the need for reliable and high-quality data transmission is a challenge. How did digital coding surfaces address this problem.

I am struggling to identifying the novelty of this paper:

1. Digital coding surface is not new;
2. Only power amplifiers are added to the system, the power amplifier was made from conventional technologies;
3. No new modulation schemes or underlying theory is presented.

As it stands, I would recommend this paper is submitted to IEEE relevant journals for more specific readers.

Reviewer #2 (Remarks to the Author):

In this paper, Wang et al. demonstrated a platform for simultaneous wireless information and power transfer (SWIPT) by using a reflection-type amplifying programmable metasurface (APM). The APM consists of amplifying unit cells that increases the maximum power of the incident. It can also dynamically modulate the phase distribution such that the energy beam and information beam are combined in a single beam for wireless charging and data transmission. As an example, a 60-degree

beam scanning coverage was demonstrated by phase modulation. Moreover, the authors showed that a joint modulation strategy by introducing continuous waveform can reduce the peak-to-average power ratio (PAPR), which can help mitigate the degradation caused by high-order digital modulation. As demos, the authors demonstrated SWIPT by 1. lighting up LEDs and transmitting 4QAM data in different directions and 2. Lighting up LEDs and transmitting real-time video. The jointly modulated APM demonstrated in this paper may inspire a new path to a future SWIPT system. The paper is well written and the data is also clearly presented; therefore, I recommended this paper for consideration of publication provided that my following questions and comments are addressed.

1. The authors argue that one significance of the APM is that it can reduce the PAPR that improves maximum transmitted power and flexibility in conveying energy and information. High PAPR may exceed the breakdown voltage of a rectifier and thus limits its conversion efficiency. It may also result in low output average power due to the saturation output power of the transmitter by the peak value. I am wondering why it is not the peak power value instead of the peak-to-average power ratio that put the limits of the maximum power transmission?

2. Following the above question, could the authors clarify in Fig. 3d and 3e, what ratio of CW and QAM signals is used in the joint modulation method? If the ratio is high, it is not a surprise to see the joint-modulation signal has a similar conversion efficiency to the CW signals because it is overwhelmed by the CW signals. I doubted the conclusion that "The joint modulation method can eliminate the effect of digital signal on the PAPR, enabling complex high-order modulation and further improving signal transmission rate and DC output power (line 178-180)". As shown in Fig. 3d, indeed the PAPR reduces by joint modulation, but at the same time, the amplitude also increases. Why is not the peak amplitude that determines the maximum DC output power?

Here below are some minor comments:

3. Could the authors provide the full name of QAM when it is shown the first time in the paper as all other abbreviations.

4. Could the authors elaborate the exact meaning of "coverage" in sentence (line 17-20) "As the basic hardware, ... by its coverage and stable phase modulation ...".

5. Regarding Fig. 2b, could the authors discuss how to choose the relative frequency difference between information and energy? Is there a concentration in design?

6. Why the metasurface is designed to convert the polarization state of the incident beam from X-polarized state to Y-polarized state? Is the legend of the blue dashed line in Fig. 4b Ex to Ex ?

7. In the discussion section, could the authors also discuss the limitations of their proposed joint-modulated APM?

Reviewer #3 (Remarks to the Author):

In this manuscript, the authors suggested and demonstrated a new way of achieving simultaneous wireless information and power transfers (SWIPT) by a joint modulation method. Assignment of two, different frequency channels for the power and information transfer has enabled the simultaneous optimization of the power and information transport. This allowed the authors to characterize their proposed system with an ultra-low high peak-to-average power ratio (PAPR). The manuscript has also been written in a good, readable fashion and I really appreciate the authors' creative approach. However, the necessity to assign another frequency channel dedicated to the power transport prevents me from considering it as a SWIPT method. It seems that the PAPR is not valid anymore as soon as there is another independent degree of freedom to get arbitrarily low PAPR. In this regard, I would recommend this manuscript for publication at Nature Communications if the authors make me convinced by addressing this fundamental question as well as other technical concerns in the following:

1. Amplification also means additional energy costs. Could the author make a quantitative model or a convincing remark to claim this method is better than the other methods in terms of the sustainability of wireless communication?

2. In line 98, could the authors refer to the current state-of-the-art?
3. I personally like the authors' idea to separate the channels for power and information transport into two distinct frequencies in a practical sense. However, I am also concerned that as soon as they split the frequency channel, it does not seem to be a SWIPT nor a single beam method. They assigned another frequency channel or beam dedicated to the power transport that, otherwise, would provide another information bandwidth. The splitting in frequency also means each receiving device (e.g., IoT devices) needs to be equipped with the ability to resolve the frequency as well, which adds another cost and system complexity.
4. Can the authors discuss more in detail what made them decide the 4-level phase to steer the beam? I think they need to discuss which aspects one can improve by increasing the number of phase steps as well as which performance would be limited other than attributing to system complexity.
5. Regarding the frequency difference between the power and information transport channels (Fig. 2b), it seems necessary for the authors to add some remarks on the effect of the frequency difference on the performance.
6. In Fig. 1b (ii), each of the four parts in the unit cell manifests dispersive properties; the phase changes as a function of frequency. Is this advantageous or should it be avoided? If the latter is the case, what makes them such dispersive and how one can relieve this for future improvements?

Responses to Reviewers' Comments

We would like to thank all reviewers for their constructive suggestions and comments, which help us improve the quality of the manuscript (NCOMMS-23-22329) significantly. We have revised the original manuscript and the supplementary material carefully according to these suggestions and comments. All main changes are marked in red fonts in the revised manuscript. Below are our item-to-item responses to the reviewers' comments.

General comments from Reviewer #1:

The concept of simultaneous wireless information and power transfers (SWIPT) is not new and it has been studied by various academic communities for a long time since 2008. There is a long list of papers can be found below:

"Simultaneous Wireless Information and Power Transfer in Modern Communication Systems" by Zhang, R., & Ho, C. K. (2013)

"Simultaneous Wireless Information and Power Transfer: A Survey of Recent Developments" by Ju, H., Zhang, R., & Guizani, M. (2014)

"Simultaneous Information and Power Transfer for Energy Harvesting Communication Systems" by Varshney, L. R. (2008)

"Simultaneous Wireless Information and Power Transfer in Cooperative Networks" by Liu, L., Zhang, R., & Chua, K. C. (2013)

"Simultaneous Wireless Information and Power Transfer in MIMO Systems" by Clerckx, B., & Oechtering, T. J. (2013)

"Simultaneous Information and Power Transfer for Broadband Wireless Systems" by Park, J. M., Zhang, R., & Fung, C. (2013)

"Simultaneous Information and Power Transfer in Multiuser MIMO Systems" by Zhou, X., Zhang, R., & Nallanathan, A. (2013)

"Simultaneous Wireless Information and Power Transfer in Cognitive Radio Systems" by Dey, S., Niyato, D., & Kim, D. I. (2013)

"Simultaneous Wireless Information and Power Transfer for MIMO-OFDM Systems" by Huang, K., Wu, D., & Zhang, R. (2014)

"Simultaneous Information and Power Transfer in Single-Antenna Interference Networks" by Tandon, A., & Motani, M. (2014)

The paper combined this concept with programmable surfaces or digital coding surfaces, it however does not address what the added value is for SWIPT. There are ongoing challenges such as improving power efficiency, which involves optimizing the energy transfer efficiency and reducing energy losses during transmission, conversion, and harvesting processes. Balancing the power requirements with the need for reliable and high-quality data transmission is a challenge. How did digital coding surfaces address this problem.

Authors Response:

We appreciate the reviewer for providing valuable comments that help us further improve the quality of the paper and outstand the novelty of this work. It is true that SWIPT has been

introduced since 2008 and many investigations have been reported¹⁻¹⁸, including the papers provided by the reviewer. However, the reported works¹⁻³ were mainly based on the MIMO technology or multiple-antenna system. The MIMO technology and multiple antenna system can improve the communication rate and capacity by using beam forming strategy or multiple-physical channels³⁻⁵, but the hardware architecture of a typical multiple-antenna transmitter requires numerous RF-link devices and digital processing devices, such as transceivers, power amplifiers, FPGAs, LNAs, and so on. Moreover, transmitting information and energy by employing broadband resources is an alternative approach⁶, but this approach takes up a lot of spectral resources and puts forward a new challenge for hardware realization.

On the contrary, the digital coding metasurface doesn't need complex and large numbers of devices, which can significantly reduce the power consumption, improve transmission efficiency, and mitigate the transmission loss. To simply compare the power consumptions and transmission efficiencies of a typical multiple-antenna transmitter and the proposed amplifying programmable metasurface (APM), we construct a schematic of a typical multiple-antenna transmitter, as shown in Fig. R1a. The main active components used in the multiple antenna transmitter are listed in Table R1. We make two cases to outstand the advantage of the proposed method, compared with the multiple-antenna transmitters in SWIPT.

Case 1: We calculate the power consumptions of the multiple-antenna transmitter and the proposed APM with the assumption that the number of antennas is equal to that of metasurface units, and the output powers from the antenna and the metasurface unit are identical. The calculated results (Fig. R1b) show that the total power consumption of the proposed APM is about one-sixth of the multiple-antenna transmitter, which demonstrates the proposed APM can significantly reduce the power consumption. We also count the number of components used in these two architectures shown in Fig. R1c. Although the proposed APM has a large number of PIN diodes and power amplifiers, it presents one-sixth of the power consumption lower than the multiple-antenna transmitter. This is primarily attributed to the significantly lower power consumption of PIN diodes (only 12.75 mW per unit) and the absence of power-hungry devices such as transceivers and FPGA. The circulators and duplexers in the multiple-antenna transmitter are lossy components, which in turn increase the power consumption and energy loss. Furthermore, the multiple-antenna transmitter incurs higher costs compared to the proposed APM due to the large use of expensive components, including FPGA and transceiver.

Fig. R1 Comparison of power consumptions between a multiple-antenna transmitter and APM. **a** Typical hardware architecture of the multiple-antenna transmitter. **b** Power consumptions of the multiple antenna transmitter and the proposed APM, where the maximum radiated power at the antenna side and the unit cell of APM are identical. **c** The numbers of various components used in the multiple antenna transmitter and APM.

Type	Model	Manufacturer
FPGA	Z-7035	AMD-Xilinx
Power amplifier	TQP7M9103	Qorvo
LNA	QPL9058	Qorvo
Transceiver	AD9371	ADI
Gain Block	TQP9062	Qorvo

Table R1. Components of multiple-antenna transmitter.

Case 2: We compare the transmission energies of the proposed APM and multiple-antenna transmitter to demonstrate the advantages of the proposed APM. Here the power consumption of the two solutions is identical for the sake of fairness. From Fig. R1b and Fig. R1c, the number of antennas in a multiple-antenna transmitter is approximately reduced to six in terms of the same power consumption. With the assumption that the same feeding power of two solutions, we can numerically calculate the electric field distribution on a specific region at 1 meter away from the boresight of the surface, as shown in Fig. R2a. And the steering angle of the two solutions is set to $\theta = 0^\circ$ for simplicity. From numerical results, we observed that the electric field intensity of the multiple-antenna transmitter is much lower than that of the APM. From the energy pattern shown in Fig. R2b, the transmission energy of the multiple-antenna transmitter is 9.1dB less than that of APM at the direction $\theta = 0^\circ$, which demonstrates that the APM can significantly increase transmission efficiency. Moreover, the APM presents lower energy loss than the multiple-antenna transmitter by steering a narrow beam. The higher transmission efficiency and lower energy loss of APM are mainly due to two reasons: one is the aperture of APM is larger than the multiple-antenna transmitter; the other is the unit cell of APM can enhance rather than attenuate the spatial electromagnetic wave. Hence, the APM provides outstanding performance to boost the energy transmission efficiency and reduce the energy transmission loss, which may facilitate the development of the SWIPT technology. Moreover, these advantages will be unsurprisingly extended as the size of APM increases.

Fig. R2 Comparison of energy transmissions between the APM and a 2×3 antenna array. **a** Electric field distribution on a $1 \text{ m} \times 1 \text{ m}$ region at 1 m away from the boresight of the APM and antenna. **b** Normalized radiated energy patterns of the antenna array and APM under the same feeding power.

We deeply agree with the reviewer that the one of challenges in practical SWIPT is balancing the power requirement with the need for reliable and high-quality data transmission. Reliable and high-quality data transmission relies on the recognizable energy of each symbol or the reasonable signal-to-noise ratio (SNR) of the transmitted signal. For a co-located receiver with one antenna in SWIPT, the conventional strategy based on the MIMO system is using the technologies of spatial multiplex, power splitting, antenna switching, and time switching to transmit wireless energy and information⁵. However, this will deteriorate the energy of the baseband signal and reduce the energy harvesting efficiency. Moreover, increasing the proportion of baseband energy will introduce a time-varying envelope with a nonnegligible peak-to-average ratio (PAPR). **To address this challenge, we introduce a new degree of freedom by exploiting a continuous wave at another frequency to significantly mitigate the impact of baseband signals and employ the proposed APM to form a high-directional beam to increase the signal energy.** We termed this modulation strategy as joint modulation in this work. Based on the joint modulation strategy, the harvesting energy can be written as $P_T = \eta(P_i, \text{PAPR}) \left(E \left[\| \mathbf{H} \mathbf{x}(n) \|^2 \right] + P_c \right)$, where $\eta(P_i, \text{PAPR})$ is the conversion efficiency which is a function of the average power and PAPR of an input signal, $E[\cdot]$ denotes statistical expectation, \mathbf{H} denotes the transmission channel from the transmitter to the receiver, $\mathbf{x}(n)$ denotes random baseband signal at n th symbol interval, and P_c is the energy of continuous wave. The quality of data transmission relies on the average energy of symbols $E \left[\| \mathbf{H} \mathbf{x}(n) \|^2 \right]$. And the harvesting energy depends on the continuous wave energy P_c . **We can adjust the power of information and continuous wave to reach the optimum energy conversion efficiency and data transmission quality in terms of different sensitivity of the energy converter and wireless information receiver.** In general, the sensitivity of an energy converter is much larger than a wireless information receiver. This allows us to provide reasonable SNR for reliable and high-quality data transmission while maintaining a high energy conversion efficiency by using a high-power continuous wave at different frequencies. Moreover, the high-directional beam generated by the APM further improves both the energy

of the continuous wave and modulated baseband signal. Here we experimentally verified the good performance of the proposed joint modulation strategy for high-quality and reliable data transmission. The experiment setup is schematized in Fig. R3a. The CW signal is configured at 4GHz and a 64QAM signal is centered at 3.98GHz. We recorded the SNR of the 64QAM signals under different average output power from the APM, as depicted in Fig. R3b. The SNR consistently exceeds 30 dB across the average output power range of -10 dBm to 12 dBm. We observed that the variation of SNR is less than 2.84 dB. The process of the experiment can be found in Supplementary Movie 3. In summary, the proposed joint modulation method can independently configure the power of information and energy to provide a high-quality data transmission with a reliable SNR as well as a high conversion efficiency.

Fig. R3 Experimental verification for high-quality data transmission. **a** Schematic view of the experimental setup. **b** SNR performance versus the average output power of the APM.

In summary, the main contribution of the proposed APM and the joint modulation method for the developments of SWIPT in this paper are listed as follows:

1. A new architecture of digital coding metasurface integrating with an amplifier into the unit cell is proposed, which increases the transmission efficiency and reduces the energy loss during the transmissions by forming a high-directional beam and enhancing the spatial wave.
2. A new joint modulation method is introduced by exploiting a CW signal to transmit the energy, which increases the energy-harvesting efficiency by reducing PAPR of the transmitted signal.
3. A new strategy is presented using the proposed joint modulation method by independently configuring the energy and information at different frequencies, which keeps high SNR for high-quality data transmission during wireless power and information transmission.
4. A new platform is established based on the amplifying programmable metasurface, which reduces the system cost by diminishing the number of costly components.

Specific Comments from Reviewer #1:

Reviewer #1 -- Comment 1:

1. Digital coding surface is not new;

Authors Response:

We thank the reviewer for this comment. Yes, the concept of digital coding metasurface is not new and numerous studies have realized various functions to illustrate its powerful ability in manipulating electromagnetic waves. **However, the proposed amplifying programmable metasurface is new for the SWIPT solution to address the challenges in the conventional multiple-antenna system and digital-coding metasurface system.** Compared with the multiple-antenna transmitter, the outstanding performance of the amplifying programmable metasurface has been discussed above and verified by Case 1 and Case 2, and the experiments. In contrast to the conventional digital coding metasurface⁷ which requires a large number of elements, the proposed amplifying programmable metasurface shows tremendous advantages in SWIPT, such as low aperture size, high-power handling ability, and spatial enhancement. Besides, we have reviewed the previous works of amplifier-based surfaces. We found that part of the state of art amplifying metasurfaces⁸⁻¹² has fixed functions in the electromagnetic wave manipulation, which can not configure the beams in a programmable way; while the other amplifier-based metasurfaces¹³⁻¹⁵ can not generate the high-directional beams dynamically for simultaneous information and energy transfers. In summary, **the proposed amplifying metasurface provides a new platform to achieve reliable data transmission and lower energy transmission loss with lower volume, cost, and power consumption.**

Reviewer #1 -- Comment 2:

2. Only power amplifiers are added to the system, the power amplifier was made from conventional technologies;

Authors Response:

We thank the reviewer for this comment. The power amplifier design in this work is different from the conventional power amplifier. The conventional power amplifier requires to design the matching network in fixed source and load situations without coupling from the external signals, which means that the impedances of the source and load are invariant. This is because the performance of amplifiers such as the gain, return loss, maximum output power, and maximum efficiency will be changed with the input and output impedance¹⁶, which is called as load-pull and source-pull effects. Hence the conventional amplifier is usually covered by a metal box for preventing the coupling from the external signals.

However, in the proposed work, the amplifier is exposed to the air and surrounded by adjacent radiating elements. Hence, we need to utilize a periodical boundary condition to design an amplifying unit cell, which has taken into account the coupling from the adjacent radiating element, as shown in Supplementary Note 4. Moreover, the source impedance of the amplifier varies with the state of the PIN diodes placed on the patch and the phase shifter, which is different from the conventional amplifier. According to the load-line theory¹⁷, we need to optimize the output impedance Z_{out} of the amplifier within the impedance region to approach the optimized impedance $Z_{opt} = V_{dc} / (I_{max} / 2)$. Here, we measured the load-pull data of the amplifier TQP7M9102 by using a thru-reflect-line (TRL) fixture, as shown in Fig. R4a. The load-pull data record the maximum output power from the amplifier at 4GHz with a given load impedance Z_L , which are a set of contours, as shown in Fig. R4b. For demonstrating the difference from the conventional method, we extracted the output impedance of the amplifier in a unit cell from the periodical boundary condition and open boundary at 4GHz. The output impedances seen from the amplifier based on two methods are plotted in the impedance locus

of the measured load-pull data, as shown in Fig. R4b. From Fig. R4b, we can see that the conventional design method presents about 1 dB deterioration in the maximum output power.

In summary, **the proposed design method is different from the conventional method since the input impedance of the amplifier varies with the reflected phase, and all impedances are designed using the periodical boundary conditions.**

Fig. R4 Measurement of load-pull data of the amplifier and impedance analysis. **a** The TRL fixture for measuring the load-pull data of the amplifier. **b** The distribution of the output impedance within the load-pull impedance region according to different design methods. The red asterisk is the impedance extracted from the periodical boundary condition. The blue asterisk is an impedance extracted from the open boundary.

Reviewer #1 -- Comment 3:

3. No new modulation schemes or underlying theory is presented.

Authors Response:

We appreciate your feedback and have carefully considered your comment regarding to the lack of new modulation schemes or underlying theories presented in our work. Compared with the modulation scheme of SWIPT by using the multiple-antenna system^{1,2}, **we have proposed a novel modulation method depicted in Fig. 3a of the manuscript.** Here, the probability of the instantaneous power of a signal is described by a complementary cumulative distribution function. It is a statistical method to analyze the instantaneous power of the transmitted signal above its average power, corresponding to the PAPR distribution. We introduce a new degree of freedom by using the diversity and orthogonality of frequency to carry the wireless energy, which can significantly mitigate the impact of the digital modulation on PAPR to improve the energy harvesting efficiency. **The underline theory of the proposed joint modulation can be expressed as follow.** The total harvested energy is a sum of baseband and continuous waves, denoted by

$$P_T = \eta(P_a, \text{PAPR}) \left(E \left[\|\mathbf{H}\mathbf{x}(n)\|^2 \right] + P_c \right) \quad (1)$$

where $\eta(P_i, \text{PAPR})$ is conversion efficiency which is a function of average power P_a and PAPR, $E[\cdot]$ denotes statical expectation, $\mathbf{H} \in \mathbb{C}^{M \times N}$ denotes the transmission channel from the transmitter to the receiver with N th antenna equipped on the receiver and M th antenna

equipped on the transmitter, $\mathbf{x}(n) \in \mathbb{C}^{N \times 1}$ denotes random baseband signal at the n th symbol interval, and P_c is the average energy of continuous wave. And the baseband transmission from the APM to the receiver can be modeled as

$$\mathbf{y}(n) = \mathbf{H}\mathbf{x}(n) + z(n) \quad (2)$$

where $\mathbf{y}(n) \in \mathbb{C}^{N \times 1}$ represents the received baseband signal at the n th symbol, and $z(n)$ denotes the receiver noise vector. Since only one physical channel in the proposed system uses a high-directional beam and one antenna equipped on the receiver ($M=1, N=1$), the total harvested energy can be written as

$$P_T = \eta(P_a, \text{PAPR}) \left(H_b(\theta, \varphi) E \left[\|\mathbf{x}(n)\|^2 \right] + H_c(\theta, \varphi) P_c \right) \quad (3)$$

where $H_b(\theta, \varphi)$ and $H_c(\theta, \varphi)$ are the transmission coefficient of the physical channel of baseband signals and continuous wave, respectively, which depends on the scattered energy by the proposed APM, and P_e is the energy of continuous wave feeding to the APM. The maximum rate can be denoted $R = \log_2 \left(1 + E \left[\|\mathbf{x}(n)\|^2 \right] / P_n \right)$ because the frequency of a continuous wave is different from the modulated baseband signals, where P_n is the average energy of the receiver noise. We suppose that the ratio of the continuous wave to the baseband signal energy is α . Hence, we can adjust α to improve the total harvested energy while keeping the baseband energy for high-quality data transmission. Furthermore, the PAPR of the total transmitted signal can be readily reduced by increasing the ratio α . According to the field calculation in equation S1 in Supplementary information, the transmission coefficient in the Fresnel Zone is proportional to the scattered energy $H(\theta, \varphi) \propto |E_y|^2$. Hence, we can adjust the code distribution on the APM to form an arbitrary high-directional beam for energy and information transmission.

The DC output power is time-average energy which is a constant value in a transmitted signal. The peak amplitude of transmitted signals only determines the peak voltage across the diode, which is not the main factor for DC output power if the peak amplitude takes little proportion of the transmitted signals. Besides, a signal with a larger PAPR drives a larger current through the diode than a CW signal. This larger current results in a higher energy loss on the series resistance. And the waveform of a larger PAPR signal may over the reverse breakdown voltage and be clipped. Therefore, a transmitted signal with a low PAPR and suitable energy level will be helpful to a converter for providing DC output power. Since DC output power is time-invariant, the maximum DC output power is identical to the DC output power. We can theoretically explain the generation of DC output power from a single-diode converter under a joint modulated signal. An ideal energy-harvesting circuit model extracted from the converter circuit (Fig. R5a) is considered to analyze the DC output power, as shown in Fig. R5b. We ignore the package and all other diode parasitic for theoretical analysis simplicity. The current through junction capacitance C_J is negligible due to that capacitance is very small. Similar to the analysis of energy-harvesting circuits¹⁸, the DC output power is a function of the time-average energy of the converted signals, which can be denoted as $P_{DC} = V_{dc}^2 / 2R_L$.

$$V_{dc} = \frac{V_{dc,d}}{1 + \frac{R_s}{R_L}} \quad (4)$$

where $V_{dc,d}$ is the voltage across the diode, R_s is the series resistance, R_L is the load resistance. For an input signal in the time T , the DC output voltage of the diode is the time-average voltage across the diode, which can be written as

$$V_{dc,d} = \frac{1}{T} \int V_d dt \quad (5)$$

The voltage across the diode under the input signal $s(t)$ is related to the threshold voltage and the transient voltage of signals $s(t)$. When the input signal is larger than the reverse breakdown voltage V_r , the voltage across the diode is $-V_r$. When the diode turns off or the input signal is less than the threshold, the diode voltage V_d is equal to the signals. When the signal is larger than the threshold, the diode voltage V_d is equal to the threshold voltage V_t . The diode voltage can be denoted as

$$V_d = \begin{cases} -V_r, & s(t) \leq -V_r \\ s(t), & -V_r < s(t) < V_t \\ V_t, & s(t) \geq V_t \end{cases} \quad (6)$$

Based on the joint modulation method, the input signal to the energy-harvesting circuit is the sum of the modulated baseband signals $x(t)e^{j\omega_m t}$ and continuous signal $A_c e^{j\omega_c t}$. Note that the angular frequency of modulated baseband ω_m is different with the continuous signal ω_c . Since we employ the APM to generate a high-directional beam for wireless energy and information transmission, the modulated signal arriving at the receiver can be expressed as

$$s(t) = \square \left(h_b x(t) e^{j\omega_m t} + h_c A_c e^{j\omega_c t} \right), \quad x(t) = A_s(t) e^{j\phi(t)} \quad (7)$$

where $h_b, h_c \in \square$ represent the equivalent complex channel, and $x(t)$ is the baseband signal. Here, we take three cases (Fig. R5c) as examples for demonstrating the DC output power performance. The peak amplitude of the three cases is normalized to the reverse breakdown voltage. Case I is the converter circuit under input modulated signals with a high PAPR. Case II is an input signal with a middle PAPR, and Case III is continuous waves. According to the characteristics of diode BAT15-03w from Infineon, the threshold voltage is set to 0.224V, the series impedance is set to 5Ω , and the reverse breakdown voltage is set to 4.2V. The envelope of three modulated signals is depicted in Fig. R5c. The PAPR distributions of signals of Case I and Case II are shown in Fig. R5b. Since any CW signal has a 0 dB PAPR value, we have not calculated the PAPR distribution of Case III in Fig. R5d. Based on the above theory, we can observe that the DC output power of the converted circuit increases as the PAPR value decreases, as depicted in Fig. R5e, which is consistent with the measured results shown in Fig. 3e of the manuscript. In summary, the proposed joint modulation technique can be modeled using the underlying theory described above, which serves as a solid foundation for achieving high-quality data transmission and high conversion efficiency.

We have inserted the abovementioned discussions on the underline theory with red fonts in the revised manuscript and Supplementary Note. We have also added Supplementary Movie 3 for demonstrating the high-quality data transmissions.

We added the load-pull analysis in Supplementary Note 4.

We added Supplementary Note 2 to demonstrate the basic theory of the proposed joint modulation method.

We also added Supplementary Note 3 to show the theoretical analysis of the energy-harvesting circuit.

Fig. R5 Theoretically analysis of the DC output power according to a single-diode converter circuit. **a** The structure of a single-diode converter circuit. **b** Theoretical model of a converter circuit. **c** The waveform of input signals with different PAPR: case I, the waveform of a signal with a high PAPR; case II, the waveform of a signal with a middle PAPR; case III, the waveform of a CW signal. **d** Statistics of the peak amplitude distribution of the signal for Case I and Case II. **e** The DC output power of a converter circuit in three cases.

1. Rui Z, Chin Keong H. MIMO Broadcasting for Simultaneous Wireless Information and Power Transfer. In: 2011 IEEE Global Telecommunications Conference - GLOBECOM 2011) (2011).
2. Rubio J, Pascual-Iserte A. Simultaneous wireless information and power transfer in multiuser MIMO systems. In: 2013 IEEE Global Communications Conference (GLOBECOM) (2013).
3. Wei Z, Yu X, Ng DWK, Schober R. Resource Allocation for Simultaneous Wireless Information and Power Transfer Systems: A Tutorial Overview. Proc. IEEE 110, 127-149 (2022).
4. Telatar E. Capacity of Multi-antenna Gaussian Channels. Eur. Trans. Telecommun. 10, 585-595 (1999).
5. Krikidis I, Timotheou S, Nikolaou S, Zheng G, Ng DWK, Schober R. Simultaneous wireless information and power transfer in modern communication systems. IEEE Commun. Mag. 52, 104-110 (2014).
6. Huang K, Larsson E. Simultaneous Information and Power Transfer for Broadband Wireless Systems. IEEE Trans. Signal Process. 61, 5972-5986 (2013).
7. Amri MM, Tran NM, Park JH, Kim DI, Choi KW. Demo: Demonstration of Reconfigurable Intelligent Surface (RIS)-assisted Simultaneous Wireless Information and Power Transfer (SWIPT). In: 2022 IEEE International Conference on Communications Workshops (ICC Workshops) (2022).

-
8. Wu L, et al. A Wideband Amplifying Reconfigurable Intelligent Surface. *IEEE Trans. Antennas Propag.* 70, 10623-10631 (2022).
 9. Dong Y, Kang W, Sima B, Wu W. Retroreflector With Polarization Isolation Based on Nonreciprocal Metasurface. *IEEE Antennas Wirel. Propag. Lett.* 21, 1940-1944 (2022).
 10. Lou T, Yang X-X, He G, Che W, Gao S. Dual-Polarized Nonreciprocal Spatial Amplification Active Metasurface. *IEEE Antennas Wirel. Propag. Lett.* 20, 1789-1793 (2021).
 11. Lončar J, Grbic A, Hrabar S. Ultrathin active polarization-selective metasurface at X-band frequencies. *Phy. Rev. B* 100, 075131 (2019).
 12. Lavigne G, Caloz C. Magnetless reflective gyrotropic spatial isolator metasurface. *New J. Phys.* 23, 075006 (2021).
 13. Ma Q, et al. Controllable and Programmable Nonreciprocity Based on Detachable Digital Coding Metasurface. *Adv. Opt. Mater.* 7, 1901285 (2019).
 14. Qiu T, Jia Y, Wang J, Cheng Q, Qu S. Controllable Reflection-Enhancement Metasurfaces via Amplification Excitation of Transistor Circuit. *IEEE Trans. Antennas Propag.* 69, 1477-1482 (2021).
 15. Taravati S, Eleftheriades GV. Full-duplex reflective beamsteering metasurface featuring magnetless nonreciprocal amplification. *Nat. Commun.* 12, 4414 (2021).
 16. Ghannouchi FM, Hashmi MS. Load-pull techniques and their applications in power amplifiers design (invited). In: 2011 IEEE Bipolar/BiCMOS Circuits and Technology Meeting) (2011).
 17. Cripps S. *RF Power Amplifiers for Wireless Communications*. Artech (2006).
 18. Valenta CR, Morys MM, Durgin GD. Theoretical Energy-Conversion Efficiency for Energy-Harvesting Circuits Under Power-Optimized Waveform Excitation. *IEEE Trans. Microwave Theory Tech.* 63, 1758-1767 (2015).

General comments from Reviewer #2:

In this paper, Wang et al. demonstrated a platform for simultaneous wireless information and power transfer (SWIPT) by using a reflection-type amplifying programmable metasurface (APM). The APM consists of amplifying unit cells that increases the maximum power of the incident. It can also dynamically modulate the phase distribution such that the energy beam and information beam are combined in a single beam for wireless charging and data transmission. As an example, a 60-degree beam scanning coverage was demonstrated by phase modulation. Moreover, the authors showed that a joint modulation strategy by introducing continuous waveform can reduce the peak-to-average power ratio (PAPR), which can help mitigate the degradation caused by high-order digital modulation. As demos, the authors demonstrated SWIPT by 1. lighting up LEDs and transmitting 4QAM data in different directions and 2. Lighting up LEDs and transmitting real-time video. The jointly modulated APM demonstrated in this paper may inspire a new path to a future SWIPT system. The paper is well written and the data is also clearly presented; therefore, I recommended this paper for consideration of publication provided that my following questions and comments are addressed.

Authors Response:

We appreciate the reviewer very much for the positive comments. The insightful comments are very constructive for further improvement of this work. In the following, we address the specific comments point-by-point whilst revising our manuscript.

Specific comments from Reviewer #2:

Reviewer #2 -- Comment 1:

1. The authors argue that one significance of the APM is that it can reduce the PAPR that improves maximum transmitted power and flexibility in conveying energy and information. High PAPR may exceed the breakdown voltage of a rectifier and thus limits its conversion efficiency. It may also result in low output average power due to the saturation output power of the transmitter by the peak value. I am wondering why it is not the peak power value instead of the peak-to-average power ratio that put the limits of the maximum power transmission?

Authors Response:

We greatly appreciate the insightful questions raised by the reviewer. In this work, we propose an APM as a transmitter to enhance the upper limits of power transmission compared to a lossy digital coding metasurface. Additionally, we introduce a joint modulation method aimed at reducing the PAPR of the transmitted signal. This modulation method, in turn, helps increase the conversion efficiency of the converter and average output power from the APM. **For instantaneous power transmission, the peak power value of the transmitted signal indeed limits the maximum power. However, wireless energy charging depends not only on the transient power transmission but also on the cumulative energy transmission.** For the power transmission over a time T , the accumulated power transmission is an integral to the transmitted power which can be denoted as $P_a = \int_T P_t dt = \int_T (|x(t)|^2 + P_c(t)) dt$, where P_t is the transient power of the transmitted signal, $x(t) \in \mathbb{C}$ denotes the envelope of the modulated baseband signal, and P_c is the average energy of continuous wave. Therefore, the peak power value of the transmitted signal for maximum accumulated power transmission can be ignored if it takes a little proportion of the time. With the assumption that the maximum transmitted power of APM is fixed, the accumulated maximum power of the transmitted signal with a high PAPR is less than that of with a low PAPR. Hence, the PAPR determines the maximum accumulated power transmission, while the peak power value dictates the instantaneous maximum power transmission.

Reviewer #2 -- Comment 2:

2. Following the above question, could the authors clarify in Fig. 3d and 3e, what ratio of CW and QAM signals is used in the joint modulation method? If the ratio is high, it is not a surprise to see the joint-modulation signal has a similar conversion efficiency to the CW signals because it is overwhelmed by the CW signals. I doubted the conclusion that “The joint modulation method can eliminate the effect of digital signal on the PAPR, enabling complex high-order modulation and further improving signal transmission rate and DC output power (line 178-180)”. As shown in Fig. 3d, indeed the PAPR reduces by joint modulation, but at the same time, the amplitude also increases. Why is not the peak amplitude that determines the maximum DC output power?

Authors Response:

We thank the reviewer for these professional questions. The ratio of CW and 64QAM signals in the joint modulation method is 30 dB in Figs. 3d and 3e. And we measured the PAPR values of all modulated signals in Fig. 3d under the same average output power 10dBm.

We apologize for this inaccurate description “the joint modulation method can eliminate the effect on the digital signal on the PAPR”. The PAPR values of signals with different modulation orders are close to zero with the increasing ratio of CW signal, but they are not equal to zero. We have corrected this inaccurate description by using “significantly mitigate” in the revised main text. Besides, the joint modulation method can improve the transmission rate and increase the DC output power under the same average output power from the APM. Since the transmission rate is proportional to the modulation order under the same symbol rate ($R \propto \log_2(M)$), we can realize a high-order modulation with ultra-low PAPR for a higher transmission rate based on the joint modulation method. In addition, the CW is orthogonal to the modulated baseband signal, resulting that the signal-to-noise ratio (SNR) of the baseband signal being independent of the CW signal, which has been verified by the experiment shown in Supplementary Movie 3. The schematic view of the experiment is shown in Fig. R6a. Here, the CW signal is configured at 4GHz and a 64QAM signal is centered at 3.98GHz. From the experiment, the variation of SNR of the information signals under different average output power from the APM is less than 2.84 dB. Therefore, we can adjust the energy of information with a higher SNR to further improve the transmission rate.

Fig. R6 Experimental verification for high-quality data transmission. **a** Schematic view of the experimental setup. **b** SNR performance versus the average output power of the APM.

The DC output power is the time-average energy which is a constant value in a transmitted signal. The peak amplitude of transmitted signals only determines the peak voltage across the diode, which is not the main factor for DC output power if the peak amplitude takes little proportion of the transmitted signals. Besides, a signal with a larger PAPR drives a larger current through the diode than a CW signal. This larger current results in a higher energy loss on the series resistance. And the waveform of a larger PAPR signal may over the reverse breakdown voltage and be clipped. Therefore, a transmitted signal with a low PAPR and suitable energy level will be helpful to a converter for providing DC output power. **Since the DC output power is time-invariant, the maximum DC output power is identical to the DC output power.** We can theoretically explain the generation of DC output power from a single-diode converter under a joint modulated signal. An ideal energy-harvesting circuit model extracted from the converter circuit (Fig. R7a) is considered to analyze the DC output power, as shown in Fig. R7b. We ignore the package and all other diode parasitic for

theoretical analysis simplicity. The current through junction capacitance C_J is negligible due to that capacitance is very small. Similar to the analysis of energy-harvesting circuits¹, the DC output power is a function of the time-average energy of the converted signals, which can be denoted as $P_{DC} = V_{dc}^2 / 2R_L$.

$$V_{dc} = \frac{V_{dc,d}}{1 + \frac{R_s}{R_L}} \quad (1)$$

where $V_{dc,d}$ is the voltage across the diode, R_s is the series resistance, R_L is the load resistance. For an input signal in the time T , the DC output voltage of the diode is the time-average voltage across the diode, which can be written as

$$V_{dc,d} = \frac{1}{T} \int_T V_d dt \quad (2)$$

The voltage across the diode under the input signal $s(t)$ is related to the threshold voltage and the transient voltage of signals $s(t)$. When the input signal is larger than the reverse breakdown voltage V_r , the voltage across the diode is $-V_r$. When the diode turns off or the input signal is less than the threshold, the diode voltage V_d is equal to the signals. When the signal is larger than the threshold, the diode voltage V_d is equal to the threshold voltage V_t . The diode voltage can be denoted as

$$V_d = \begin{cases} -V_r, & s(t) \leq -V_r \\ s(t), & -V_r < s(t) < V_t \\ V_t, & s(t) \geq V_t \end{cases} \quad (3)$$

Based on the joint modulation method, the input signal to the energy-harvesting circuit is the sum of the modulated baseband signals $x(t)e^{j\omega_m t}$ and continuous signal $A_c e^{j\omega_c t}$. Note that the angular frequency of modulated baseband ω_m is different with the continuous signal ω_c . Since we employ the APM to generate a high-directional beam for wireless energy and information transmission, the modulated signal arriving at the receiver can be expressed as

$$s(t) = \square \left(h_b x(t) e^{j\omega_m t} + h_c A_c e^{j\omega_c t} \right), \quad x(t) = A_s(t) e^{j\phi(t)} \quad (4)$$

where $h_b, h_c \in \square$ represent the equivalent complex channel, and $x(t)$ is the baseband signal. Here, we take three cases (Fig. R7c) as examples for demonstrating the DC output power performance. The peak amplitude of the three cases is normalized to the reverse breakdown voltage. Case I is the converter circuit under input modulated signals with a high PAPR. Case II is an input signal with a middle PAPR, and Case III is continuous waves. According to the characteristics of diode BAT15-03w from Infineon, the threshold voltage is set to 0.224V, the series impedance is set to 5Ω , and the reverse breakdown voltage is set to 4.2V. The envelope of three modulated signals is depicted in Fig. R7c. The PAPR distributions of signals of Case I and Case II are shown in Fig. R7b. Since any CW signal has a zero PAPR value, we have not calculated the PAPR distribution of Case III in Fig. R7d. Based on the above theory, we can observe that the DC output power of the converted circuit increases as the PAPR value decreases, as depicted in Fig. R7e, which is similar to the measured results shown in Fig. 3e of the manuscript.

Fig. R7 Theoretically analysis of the DC output power according to a single-diode converter circuit. **a** The structure of a single-diode converter circuit. **b** Theoretical model of a converter circuit. **c** The waveform of input signals with different PAPR: case I, the waveform of a signal with a high PAPR; case II, the waveform of a signal with a middle PAPR; case III, the waveform of a CW signal. **d** Statistics of the peak amplitude distribution of the signal for Case I and Case II. **e** The DC output power of a converter circuit in three cases.

We have inserted the discussion above with red fonts in the revised manuscript.

We have added Supplementary Note 3 to show the theoretical analysis of the energy-harvesting circuit.

We have added Supplementary Movie 3 to demonstrate that the SNR of the information is independent of the energy of the CW signal.

1. Valenta CR, Morys MM, Durgin GD. Theoretical Energy-Conversion Efficiency for Energy-Harvesting Circuits Under Power-Optimized Waveform Excitation. *IEEE Trans. Microwave Theory Tech.* **63**, 1758-1767 (2015).

Reviewer #2 -- Comment 3:

3. Could the authors provide the full name of QAM when it is shown the first time in the paper as all other abbreviations.

Authors Response:

Thank you very much for this valuable comment. We have added the full name (Quadrature Amplitude Modulation) of QAM at its first-mentioned place with red fonts in the revised manuscript.

Reviewer #2 -- Comment 4:

4. Could the authors elaborate the exact meaning of “coverage” in sentence (line 17-20) “As the basic hardware, ... by its coverage and stable phase modulation ...”.

Authors Response:

Thank you very much for this valuable comment. The coverage in SWIPT applications means the maximum power transmission distance from the digital coding metasurface. We have added “(the maximum power transmission distance from a digital coding metasurface)” with red fonts in the revised manuscript.

Reviewer #2 -- Comment 5:

5. Regarding Fig. 2b, could the authors discuss how to choose the relative frequency difference between information and energy? Is there a concentration in design?

Authors Response:

Thank you very much for this professional and valuable comment. The relative frequency difference between information and energy primarily depends on the PAPR of the transmitted signals, the quality of the data transmission, and the beam performance. According to the joint modulation method, the PAPR can be manipulated by adjusting the ratio of CW signal to the information signal. The amplitudes of CW and information can be independently controlled at different frequencies, which is independent of the relative frequency difference. In terms of data transmission quality, the proposed APM can maintain high-quality data transmission because it enhances the signal energy in the operating frequency band instead of attenuating it. To assess the beam performance of the APM, we measured the reflected beam at the APM’s azimuth in the microwave chamber. The measured steering angle of APM in the operating band (ranging from 3.95GHz to 4.05GHz) indicates a maximum difference of 3° in steering direction, as depicted in Fig. R8a. Additionally, the APM’s half-power beamwidth (HPBW) in the operating band exhibited a maximum deviation of 1.78° , as illustrated in Fig. R8b. The deviation observed in the HPBW and steering direction indicates a high degree of beam convergence. Furthermore, in the operating band, at 1 meter away from APM, the maximum difference in the maximum reflected power at different frequencies is merely 0.71 dB, as shown in Fig. R8c. The slight variation in beam performance can be attributed to the approximate 1.87 mm wavelength difference in the free space between the lower and upper frequencies, which accounts for 2.5% of the wavelength at 4GHz. **Therefore, the proposed APM can provide a highly reliable physical channel for transmitting information and energy in this work.**

Fig. R8 The performance of the steering beams of the APM in the operating band. **a** Measured steering angles of the APM at different. **b** Measured HPBW of APM at different frequencies. **c** Measured saturation power of APM at different frequencies, and receiving antenna placed at 1 meter away from the APM.

We have inserted the discussion on the configuration of the relative frequency between energy and information in the revised manuscript by using red fonts. We added the discussion on the relative frequency difference in the discussion part of the revised manuscript. We also added Supplementary Note 7 for indicating the beam performance at the operating band.

Reviewer #2 -- Comment 6:

6. Why the metasurface is designed to convert the polarization state of the incident beam from X-polarized state to Y-polarized state? Is the legend of the blue dashed line in Fig. 4b?

Authors Response:

Thank you very much for these professional questions. There are two reasons for converting the polarization of the incident wave. One is to reduce the interference of multi-path from the feeding antenna to the receiving antenna for improving the quality of data transmission, as schematized in Fig. R9. Since the receiving antenna is orthogonal to the feeding antenna, the leakage signals received from the feeding antenna can be ignored. The other one is to improve the isolation from the reradiating patch to the receiving patch in a unit cell of the APM for making sure that the APM operates at a stable region. The stability problems have been discussed in Supplementary Note 4. The legend of the blue dashed line in Fig.4b represents the reflection of the APM under the scenario where a y-polarized wave is incident on the APM from the feeding antenna, and the x-polarized receiving antenna captures the x-polarized waves. So the blue dashed line denotes E_y to E_x .

Fig. R9 Multi-path transmission channels of the APM system.

Reviewer #2 -- Comment 7:

7. In the discussion section, could the authors also discuss the limitations of their proposed joint-modulated APM?

Authors Response:

Thank you very much for the valuable comment. The benefits of the proposed joint-modulated APM for SWIPT are limited by the performance of APM, such as the level of enhancement, the operating bandwidth, and the maximum output power from APM. To address these limitations, future improvements could involve implementing a wideband power amplifier and patch scatterer to enhance the operating band. Additionally, utilizing an amplifier with higher gain and saturation output power or enlarging the aperture of the metasurface would prove beneficial in increasing the maximum output power of APM, thereby extending the range for wireless charging and information transmission. We have added these limitations in red fonts in the discussion part of the revised manuscript.

General comments from Reviewer #3:

In this manuscript, the authors suggested and demonstrated a new way of achieving simultaneous wireless information and power transfers (SWIPT) by a joint modulation method. Assignment of two, different frequency channels for the power and information transfer has enabled the simultaneous optimization of the power and information transport. This allowed the authors to characterize their proposed system with an ultra-low high peak-to-average power ratio (PAPR). The manuscript has also been written in a good, readable fashion and I really appreciate the authors' creative approach. However, the necessity to assign another frequency channel dedicated to the power transport prevents me from considering it as a SWIPT method. It seems that the PAPR is not valid anymore as soon as there is another independent degree of freedom to get arbitrarily low PAPR.

Authors Response:

Thank you for your insightful and valuable comments. The comments are very constructive for further improvement of this work. In the following, we address the specific comments point-by-point whilst revising our manuscript.

Reviewer #3 -- Comment 1:

1. Amplification also means additional energy costs. Could the author make a quantitative model or a convincing remark to claim this method is better than the other methods in terms of the sustainability of wireless communication?

Authors Response:

Thank you very much for this professional comment. Yes, a lot of works have been reported from the theoretical aspect, however, these works¹⁻³ were mainly based on the MIMO technology or multiple-antenna system. The MIMO technology and multiple antenna system can improve the communication rate and capacity by using beam forming strategy or multiple-

physical channels³⁻⁵, but the hardware architecture of a typical multiple-antenna transmitter requires numerous RF-link devices and digital processing devices, such as transceivers, power amplifiers, FPGAs, LNAs, and so on. Moreover, transmitting information and energy by employing broadband resources is an alternative approach⁶, but this approach takes up a lot of spectral resources and requires much more power consumptions by equipping more RF-link devices.

On the contrary, the proposed method doesn't need complex and large numbers of RF and digital devices, which can significantly reduce the power consumptions, improve the transmission efficiency, and mitigate the transmission loss. To simply compare the power consumption and transmission energy of a typical multiple-antenna transmitter and amplifying programmable metasurface (APM), we construct a schematic of a typical multiple-antenna transmitter, as shown in Fig. R10a. The main active components used in the multiple-antenna transmitter are listed in Table R2. We make two cases to outstand the advantage of the proposed method for sustainable wireless communications.

Fig. R10 Comparison of power consumption between a multiple-antenna transmitter and APM. **a** Typical hardware architecture of multiple-antenna transmitter. **b** The power consumption of the multiple antenna transmitter and the proposed APM, where the maximum radiated power at the antenna side and the unit cell of APM are identical. **c** The number of various components used in the multiple-antenna transmitter and APM.

Case 1: We calculate the power consumptions of the multiple-antenna transmitter and the proposed APM with the assumption that the number of antenna is equal to that of the APM units, and the output power from the antenna is identical to that of the APM unit. The calculated results (Fig. R10b) show that the total power consumption of the proposed APM is about **one-sixth** of the multiple-antenna transmitter, which demonstrates the proposed APM can significantly reduce the power consumption. We also count the number of components used in these two architectures shown in Fig. R10c, although the proposed APM has a large number of PIN diodes and power amplifiers but presents one-sixth of the power consumption lower than the multiple-antenna transmitter. This is primarily attributed to the significantly lower power consumption of PIN diodes (only 12.75 mW per unit) and the absence of power-hungry devices such as transceivers and FPGA. And the circulators and duplexers in the multiple-antenna transmitter are lossy components, which in turn increases the power consumption and energy loss.

Type	Model	Manufacturer
FPGA	Z-7035	AMD-Xilinx
Power amplifier	TQP7M9103	Qorvo
LNA	QPL9058	Qorvo
Transceiver	AD9371	ADI
Gain Block	TQP9062	Qorvo

Table R2. Components of multiple-antenna transmitter

Case 2: We compare the transmission energies of the proposed APM and multiple-antenna transmitter for demonstrating the advantages of the proposed APM. The power consumptions of the two solutions are supposed to be identical for the sake of fairness. From Fig. R10b and Fig. R10c, the number of antennas in a multiple-antenna transmitter is approximately reduced to six in terms of the same power consumption. With the assumption that the same feeding power of two solutions, we can numerically calculate the electric field distribution on a specific region at 1 meter away from the boresight of the surface, as shown in Fig. R11a. And the steering angle of the two solutions is set to $\theta = 0^\circ$ for simplicity. From numerical calculation results, we observed that the electric field intensity of the multiple-antenna transmitter is much lower than the APM. From the energy pattern shown in Fig. R11b, the transmission energy of the multiple-antenna transmitter is 9.1dB less than APM at the direction $\theta = 0^\circ$, which demonstrates that the APM can significantly increase transmission efficiency. Moreover, the APM presents lower energy loss than the multiple-antenna transmitter by steering a narrow beam. The higher transmission efficiency and lower energy loss of APM are mainly due to two reasons: one is the aperture of APM is larger than the multiple-antenna transmitter; the other is the unit cell of APM can enhance rather than attenuate the spatial electromagnetic wave. Hence, the APM provides outstanding performance to boost energy transmission efficiency and reduce energy transmission loss, which may facilitate the development of SWIPT technology. Moreover, these advantages will be unsurprisingly extended as the size of APM increases.

Fig. R11 Comparison of energy transmission between the APM and a 3×2 antenna array. **a** Electric field distribution on a $1 \text{ m} \times 1 \text{ m}$ region at 1 m away from the boresight of the APM and antenna. **b** Normalized radiated energy pattern of the antenna array and APM under the same feeding power.

1. Rui Z, Chin Keong H. MIMO Broadcasting for Simultaneous Wireless Information and Power Transfer. In: *2011 IEEE Global Telecommunications Conference - GLOBECOM 2011* (2011).
2. Rubio J, Pascual-Iserte A. Simultaneous wireless information and power transfer in multiuser MIMO systems. In: *2013 IEEE Global Communications Conference (GLOBECOM)* (2013).
3. Wei Z, Yu X, Ng DWK, Schober R. Resource Allocation for Simultaneous Wireless Information and Power Transfer Systems: A Tutorial Overview. *Proc. IEEE* **110**, 127-149 (2022).
4. Telatar E. Capacity of Multi-antenna Gaussian Channels. *Eur. Trans. Telecommun.* **10**, 585-595 (1999).
5. Krikidis I, Timotheou S, Nikolaou S, Zheng G, Ng DWK, Schober R. Simultaneous wireless information and power transfer in modern communication systems. *IEEE Commun. Mag.* **52**, 104-110 (2014).
6. Huang K, Larsson E. Simultaneous Information and Power Transfer for Broadband Wireless Systems. *IEEE Trans. Signal Process.* **61**, 5972-5986 (2013).

Reviewer #3 -- Comment 2:

2. In line 98, could the authors refer to the current state-of-the-art?

Authors Response:

Thank you very much for this comment. We have added the recent works of programmable metasurfaces and amplifier-based metasurfaces as references in the revised manuscript. The added previous works of programmable metasurfaces¹⁻⁴ and amplifier-based metasurfaces⁵⁻¹² are listed below:

1. Han J, *et al.* Adaptively Smart Wireless Power Transfer Using 2-Bit Programmable Metasurface. *IEEE Trans. Ind. Electron.* **69**, 8524-8534 (2022).
2. Amri MM, Tran NM, Park JH, Kim DI, Choi KW. Demo: Demonstration of Reconfigurable Intelligent Surface (RIS)-assisted Simultaneous Wireless Information and Power Transfer (SWIPT). In: *2022 IEEE International Conference on Communications Workshops (ICC Workshops)* (2022).
3. Lu H, *et al.* Eye accommodation-inspired neuro-metasurface focusing. *Nat. Commun.* **14**, 3301 (2023).

-
4. Tran NM, Amri MM, Park JH, Kim DI, Choi KW. Reconfigurable-Intelligent-Surface-Aided Wireless Power Transfer Systems: Analysis and Implementation. *IEEE Internet Things J.* **9**, 21338-21356 (2022).
 5. Zhang Z, *et al.* Active RIS vs. Passive RIS: Which Will Prevail in 6G? *IEEE Trans. Commun.* **71**, 1707-1725 (2022).
 6. Rao J, Zhang Y, Tang S, Li Z, Chiu C-Y, Murch R. An Active Reconfigurable Intelligent Surface Utilizing Phase-Reconfigurable Reflection Amplifiers. *IEEE Trans. Microwave Theory Tech.*, 1-14 (2023).
 7. Taravati S, Eleftheriades GV. Full-duplex reflective beamsteering metasurface featuring magnetless nonreciprocal amplification. *Nat. Commun.* **12**, 4414 (2021).
 8. Gao X, Yang WL, Ma HF, Cheng Q, Yu XH, Cui TJ. A Reconfigurable Broadband Polarization Converter Based on an Active Metasurface. *IEEE Trans. Antennas Propag.* **66**, 6086-6095 (2018).
 9. Wu L, *et al.* A Wideband Amplifying Reconfigurable Intelligent Surface. *IEEE Trans. Antennas Propag.* **70**, 10623-10631 (2022).
 10. Lou T, Yang X-X, He G, Che W, Gao S. Dual-Polarized Nonreciprocal Spatial Amplification Active Metasurface. *IEEE Antennas Wirel. Propag. Lett.* **20**, 1789-1793 (2021).
 11. Lončar J, Grbic A, Hrbar S. Ultrathin active polarization-selective metasurface at X-band frequencies. *Phy. Rev. B* **100**, 075131 (2019).
 12. Qiu T, Jia Y, Wang J, Cheng Q, Qu S. Controllable Reflection-Enhancement Metasurfaces via Amplification Excitation of Transistor Circuit. *IEEE Trans. Antennas Propag.* **69**, 1477-1482 (2021).

Reviewer #3 -- Comment 3:

3. I personally like the authors' idea to separate the channels for power and information transport into two distinct frequencies in a practical sense. However, I am also concerned that as soon as they split the frequency channel, it does not seem to be a SWIPT nor a single beam method. They assigned another frequency channel or beam dedicated to the power transport that, otherwise, would provide another information bandwidth. The splitting in frequency also means each receiving device (e.g., IoT devices) needs to be equipped with the ability to resolve the frequency as well, which adds another cost and system complexity.

Authors Response:

Thank you very much for your professional comment. **Exploiting frequency diversity to wirelessly transmit information and energy^{1,2} is a kind of SWIPT technology. And the proposed APM can generate a highly concentrated beam similar to a single beam with little deviation in beam direction and half-power beamwidth (HPBW) as well as the maximum output power at the operating band.** To assess the beam performance of the APM, we measured the reflected beam at the APM's azimuth in the microwave chamber. The measured steering angle of the APM within the operating band (ranging from 3.95GHz to 4.05GHz) indicates a maximum difference of 3° in steering direction, as depicted in Fig. R12a. Additionally, the APM's HPBW within the operating band exhibited a maximum deviation of 1.78°, as illustrated in Fig. R12b. The deviation observed in the HPBW and steering direction indicates a high degree of beam convergence. Furthermore, in the operating band, at 1 meter away from the APM, the maximum difference in the maximum reflected power at different frequencies was merely 0.71 dB, as shown in Fig. R12c. The slight variation in beam performance can be attributed to the approximate 1.87 mm wavelength difference in free space between the lower and upper frequencies, which accounts for 2.5% of the wavelength at

4GHz. Hence, the modulated data and energy can be conveyed in the same physical channel to the end user. Additionally, the power tone occupies only 1 Hz bandwidth, which can be negligible compared with the bandwidth of the information. For instance, the bandwidth of a 4QAM modulated information used to transmit video in the proposed SWIPT system is 2MHz which is much larger than the energy signal. We deeply agree that the splitting in frequency may result in a complex receiver by employing multiple antennas, different information decoders, and energy-harvesting circuits. However, this assumption is valid under the condition that the transmitted signals are capable of utilizing a significantly extensive spectrum resource. The proposed APM is a kind of narrow-band transmitter whose operating bandwidth is only 100MHz.

Therefore, the energy-harvesting circuit and information decoder can share one antenna and one coupler, which does not need any extra devices and costs.

Fig. R12 The performance of the steering beams of the APM in the operating band. **a** Measured steering angles of the APM at different. **b** Measured HPBW of APM at different frequencies. **c** Measured saturation power of APM at different frequencies, and receiving antenna placed at 1 meter away from the APM.

1. Huang K, Larsson E. Simultaneous Information and Power Transfer for Broadband Wireless Systems. *IEEE Trans. Signal Process.* **61**, 5972-5986 (2013).
2. An H, Park H. Energy-Balancing Resource Allocation for Wireless Cooperative IoT Networks With SWIPT. *IEEE Internet Things J.* **9**, 12258-12271 (2022).

We have changed the “single beam” to “highly concentrated beam similar to a single beam” in the discussion of the revised manuscript for accuracy.

We also added Supplementary Note 7 for indicating the beam performance in the operating band.

Reviewer #3 -- Comment 4:

4. Can the authors discuss more in detail what made them decide the 4-level phase to steer the beam? I think they need to discuss which aspects one can improve by increasing the number of phase steps as well as which performance would be limited other than attributing to system complexity.

Authors Response:

Thank you very much for this valuable comment. The 4-level phase is the optimum trade-off solution between the energy loss of the steering beam caused by phase quantization and the

power consumption as well as the complexity and cost of the hardware. The 2-level phase corresponding to the 1-bit manipulation has fewer numbers of PIN diodes and lower power consumptions as well as biasing complexity compared with high-bit solutions. However, 1-bit manipulation presents about 3dB energy loss in the steering beam, discussed in the previous works¹⁻³, which is not suitable for the wireless power transfer. 3-bit phase quantization requires much more components to control, which increases the DC power consumption, costs, and complexity of the biasing network. Therefore, we choose 2-bit phase quantization to cover the requirements of lower energy loss during transmission as well as lower power consumption and biasing complexity. As proof of the concept, based on the PWAS method in Supplementary Note 1, we numerically calculated the energy intensity of four solutions with different phase states at different steering directions at 4GHz, including 1-bit, 2-bit, 3-bit, and continuous phase. For evaluating the energy loss of phase quantization, the energy intensity of the steered beam is normalized to the continuous phase. An amplifier-based metasurface consisting 9×9 unit cells with the same aperture of the designed APM is employed in the calculation. We captured the distribution of the energy at 3 meters away from the metasurface, as shown in Fig. R13a. We can observe that the intensity increases as the phase-level increase. And from the calculated results shown in Fig. R13b, compared with the continuous phase solution, the 1-bit quantization solution shows -2.78 dB, -3.153 dB, and -3.57 dB deterioration at 0° , 15° , and 30° steering directions, respectively. While the intensity of the 2-bit quantization solution presents -0.7 dB, -0.688 dB, and -0.576 dB deterioration in three directions compared with the continuous phase, which are much lower than those of the 1-bit quantization.

Fig. R13 Normalized intensity of the electric field. **a** The magnitude of the reflected electric field under four quantized states of phase (1bit, 2bit, 3bit, continuous), which is normalized to the one with the continuous phase. **b** The intensity of the reflected electric field in three directions.

1. Zhang L, *et al.* Dynamically Realizing Arbitrary Multi-Bit Programmable Phases Using a 2-Bit Time-Domain Coding Metasurface. *IEEE Trans. Antennas Propag.* **68**, 2984-2992 (2020).
2. Wu B, Sutinjo A, Potter ME, Okoniewski M. On the Selection of the Number of Bits to Control a Dynamic Digital MEMS Reflectarray. *IEEE Antennas Wirel. Propag. Lett.* **7**, 183-186 (2008).
3. Han J, *et al.* Adaptively Smart Wireless Power Transfer Using 2-Bit Programmable Metasurface. *IEEE Trans. Ind. Electron.* **69**, 8524-8534 (2022).

We have inserted the contents of the reason for choosing phase level with red fonts in the revised main text. We also added the abovementioned analysis of phase quantization to Supplementary Note 8.

Reviewer #3 -- Comment 5:

5. Regarding the frequency difference between the power and information transport channels (Fig. 2b), it seems necessary for the authors to add some remarks on the effect of the frequency difference on the performance.

Authors Response:

Thank you very much for this valuable comment. We have added the discussion of the effect of the frequency difference on the performance in the revised main text with red fonts. The inserted discussion is as below:

“It is worth noting that the relative frequency difference between the energy and information can be flexibly configured in the operating band (from 3.95GHz to 4.05GHz). The relative frequency difference between information and energy primarily depends on the PAPR of the transmitted signals, the quality of the data transmission, and the beam performance. According to the joint modulation method, the PAPR can be manipulated by adjusting the ratio of CW signal to the information signal. The amplitude of CW and information can be independently controlled at different frequencies, which is independent of the relative frequency difference. In terms of data transmission quality, the proposed APM can maintain high-quality data transmission because it enhances the signal energy in the operating frequency band instead of attenuating it. Additionally, measurement results show that the deviations in maximum output power, half-power beamwidth, and beam direction are 0.71dB, 1.78°, and 3°, respectively, which guarantees flexibility for energy and information configuration on frequency. The deviation in beam performance is detailed in Supplementary Note 7.”

We also added the discussion on the relative frequency difference in the discussion part of the revised manuscript.

“By leveraging the independent modulation of energy and information, as well as a stable physical channel with minimal spatial variation, it becomes feasible to flexibly configure the frequency of both energy and information within the operating band. A stable physical channel within the operating band is guaranteed by the generation of a highly concentrated beam similar to a single beam by APM (see Supplementary Note 7).”

Reviewer #3 -- Comment 6:

6. In Fig. 1b (ii), each of the four parts in the unit cell manifests dispersive properties; the phase changes as a function of frequency. Is this advantageous or should it be avoided? If the latter is the case, what makes them such dispersive and how one can relieve this for future improvements?

Authors Response:

Thank you very much for this professional comment. Indeed, the absolute reflected phase presents dispersive properties during the operating band. **However, the anomalous reflection or adjustable reflection of a metasurface is primarily due to its phase difference on the**

surface, as discussed in the generalized laws of reflection and refraction¹. To adjust the steering direction of the beams from the APM, generally, we need to manipulate the reflection phase of each unit cell in terms of the predefined phase distribution. The predefined phase distribution takes the phase of state 1 as a reference. Hence, the dispersive properties of the phase difference indicate the metasurface's dispersive performance. From the simulated results of the phase difference (Fig. R14), the deviation of phase difference is only 4.88%, 3.55%, and 1.6% for 90°, 180°, and 270°, respectively, which can be ignored in manipulating the reflected beams.

Fig. R14 The phase difference of a unit cell at the operating band.

1. Yu N, *et al.* Light propagation with phase discontinuities: generalized laws of reflection and refraction. *Science* **334**, 333-337 (2011).

REVIEWERS' COMMENTS

Reviewer #2 (Remarks to the Author):

Thanks for authors' point-to-point reply to my comments. The revised manuscript has improved with:
1. Explanation of the effect of PAPR on maximum power transmission; 2. Theoretical model of joint modulation strategy; 3. Discussion on the relative frequency difference between information and energy channels; 4, Discussion of the work limitation and future research direction. Therefore, I recommend the manuscript to be considered for publication at Nature Communications.

Reviewer #3 (Remarks to the Author):

I have checked all the authors' responses and am now convinced that it is publishable to Nature Communications. Further review is not needed.

Responses to Reviewers' Comments

We would like to thank all reviewers and editors for their constructive suggestions and comments, which will help us improve the quality of the manuscript (NCOMMS-23-22329A) significantly. We have revised the original manuscript and the supplementary material carefully according to these suggestions and comments. All main changes are marked in red fonts in the revised manuscript. Below are our item-to-item responses to the reviewers' comments.

Comments from all Reviewer #2 and Reviewer #3:

Reviewer #2:

Thanks for authors' point-to-point reply to my comments. The revised manuscript has improved with: 1. Explanation of the effect of PAPR on maximum power transmission; 2. Theoretical model of joint modulation strategy; 3. Discussion on the relative frequency difference between information and energy channels; 4, Discussion of the work limitation and future research direction. Therefore, I recommend the manuscript to be considered for publication at Nature Communications.

Reviewer #3:

I have checked all the authors' responses and am now convinced that it is publishable to Nature Communications. Further review is not needed.

Authors Response:

We thank the reviewers very much for recommending publication. Here no more comments from reviewers need to be replied to.